

# Estimating probabilistic context-free grammars for proteins using contact map constraints

Witold Dyrka[1], Mateusz Pyzik[1], François Coste[2] and Hugo Talibart[2]

[1] Wydział Podstawowych Problemów Techniki, Katedra Inżynierii Biomedycznej, Politechnika Wrocławska, Wrocław, Poland
[2] Univ Rennes, Inria, CNRS, IRISA, Rennes, France

## ABSTRACT

Interactions between amino acids that are close in the spatial structure, but not necessarily in the sequence, play important structural and functional roles in proteins. These non-local interactions ought to be taken into account when modeling collections of proteins. Yet the most popular representations of sets of related protein sequences remain the profile Hidden Markov Models. By modeling independently the distributions of the conserved columns from an underlying multiple sequence alignment of the proteins, these models are unable to capture dependencies between the protein residues. Non-local interactions can be represented by using more expressive grammatical models. However, learning such grammars is difficult. In this work, we propose to use information on protein contacts to facilitate the training of probabilistic context-free grammars representing families of protein sequences. We develop the theory behind the introduction of contact constraints in maximum-likelihood and contrastive estimation schemes and implement it in a machine learning framework for protein grammars. The proposed framework is tested on samples of protein motifs in comparison with learning without contact constraints. The evaluation shows high fidelity of grammatical descriptors to protein structures and improved precision in recognizing sequences. Finally, we present an example of using our method in a practical setting and demonstrate its potential beyond the current state of the art by creating a grammatical model of a meta-family of protein motifs. We conclude that the current piece of research is a significant step towards more flexible and accurate modeling of collections of protein sequences. The software package is made available to the community.

# INTRODUCTION

## Grammatical modeling of proteins

The essential biopolymers of life, nucleic acids and proteins, share the basic characteristic of the languages: an enormous number of sequences can be expressed with a finite number of monomers. In the case of proteins, merely 20 amino acid species (letters) build millions of sequences (words or sentences) folded in thousands of different spatial structures

Corresponding author
Witold Dyrka,
witold.dyrka@pwr.edu.pl

playing various functions in living organisms (semantics). Physically, the protein sequence is a chain of amino acids linked by peptide bonds. The physicochemical properties of amino acids and their interactions across different parts of the sequence define its spatial structure, which in turn determines biological function to a great extent. Similarly to words in natural languages, protein sequences may be ambiguous (the same amino acid sequence folds into different structures depending on the environment), and often include non-local dependencies and recursive structures (*Searls, 2013*).

Not surprisingly the concept of *protein language* dates back to at least the 1960s (*Pawlak, 1965*), and since early applied works in the 1980s (*Brendel & Busse, 1984*; *Jiménez-Montaño, 1984*), formal grammatical models have gradually gained importance in bioinformatics (*Searls, 2002*; *Searls, 2013*; *Coste, 2016*). Most notably, profile Hidden Markov Models (HMM), which are weakly equivalent to a subclass of probabilistic regular grammars, became the main tool of protein sequence analysis. Profile HMMs are commonly used for defining protein families (*Sonnhammer et al., 1998*; *Finn et al., 2016*) and for searching similar sequences (*Eddy, 1998*; *Eddy, 2011*; *Soeding, 2005*; *Remmert et al., 2012*). The architecture of a profile HMM corresponds to the underlying multiple sequence alignment (MSA). Thus, the model perfectly suits modeling single-point mutations and supports insertions and deletions, but cannot account for interdependence between positions in the MSA. Pairwise correlations in a MSA can be statistically modeled by a Potts model (a type of Markov Random Field or, more generally, of an undirected graphical model). This has been highly successful to predict 3D contact between residues of a protein (*Hopf et al., 2017*), but computing the probability of new (unaligned) sequences with such a model is untractable (*Lathrop, 1994*). An alternative to MSA-based modeling, is to use formal grammars. Protomata (*Coste & Kerbellec, 2006*; *Bretaudeau et al., 2012*) are probabilistic regular models that can capture local dependencies for the characterization of protein families. Yet, as regular models, they are not well suited to capture the interactions occurring between amino acids which are distant in sequence but close in the spatial structure of the protein. In that case, formal grammars beyond the regular level are needed. Specifically, the context-free (CF) grammars are able to represent interactions producing nested and branched dependencies (an example is given in Fig. 1), while the context-sensitive (CS) grammars can also represent overlapping and crossing dependencies (*Searls, 2013*). The sequence recognition problem is untractable for CS grammars, but it is polynomial for CF and *mildly* context-sensitive grammars (*Joshi, Shanker & Weir, 1990*). However, grammatical models beyond the regular level have been rather scarcely applied to protein analysis (a comprehensive list of references can be found in *Dyrka, Nebel & Kotulska (2013)*. This is in contrast to RNA modeling, where CF grammatical frameworks are well-developed and power some of the most successful tools (*Sakakibara et al., 1993*; *Eddy & Durbin, 1994*; *Knudsen & Hein, 1999*; *Sükösd et al., 2012*).

One difficulty with modeling proteins is that interactions between amino acids are often less specific and more *collective* in comparison to RNA. Moreover, the larger alphabet made of 20 amino acid species instead of just four bases in nucleic acids, combined with high computational complexity of CF and CS grammars, impedes inference, which may lead

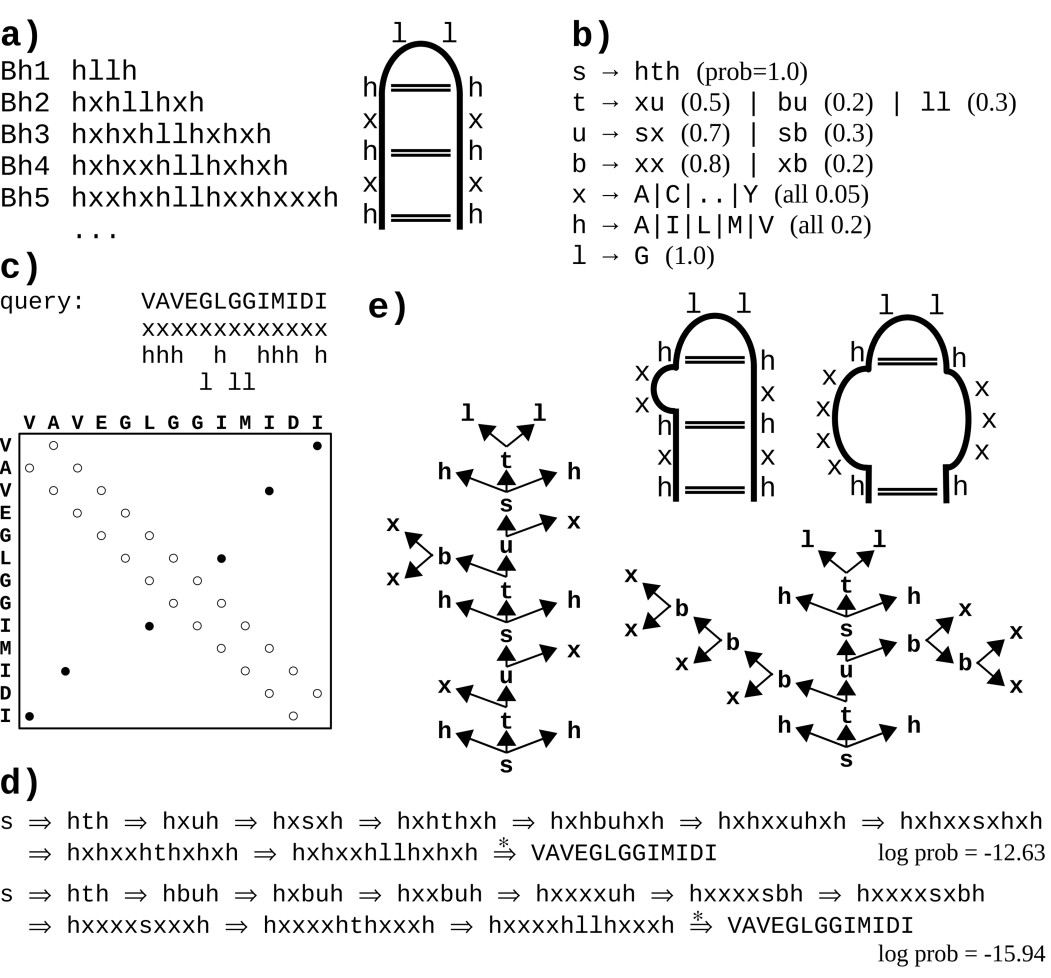

**a)**
```
Bh1 hllh
Bh2 hxhllhxh
Bh3 hxhxhllhxhxh
Bh4 hxhxxhllhxhxh
Bh5 hxxhxhllhxxhxxxh
    ...
```

**b)**
```
s → hth (prob=1.0)
t → xu (0.5) | bu (0.2) | ll (0.3)
u → sx (0.7) | sb (0.3)
b → xx (0.8) | xb (0.2)
x → A|C|..|Y (all 0.05)
h → A|I|L|M|V (all 0.2)
l → G (1.0)
```

**c)**
```
query:      VAVEGLGGIMIDI
            xxxxxxxxxxxx
            hhh  h  hhh h
               l ll
```

**d)**

s ⇒ hth ⇒ hxuh ⇒ hxsxh ⇒ hxhthxh ⇒ hxhbuhxh ⇒ hxhxxuhxh ⇒ hxhxxsxhxh
  ⇒ hxhxxhthxhxh ⇒ hxhxxhllhxhxh ⇒* VAVEGLGGIMIDI          log prob = -12.63

s ⇒ hth ⇒ hbuh ⇒ hxbuh ⇒ hxxbuh ⇒ hxxxxuh ⇒ hxxxxsbh ⇒ hxxxxsxbh
  ⇒ hxxxxsxxxh ⇒ hxxxxhthxxxh ⇒ hxxxxhllhxxxh ⇒* VAVEGLGGIMIDI
                                                          log prob = -15.94

**Figure 1** **A toy example of application of the probabilistic CFG to protein sequences.** (A) Fictitious subfamily of beta-hairpins (*Milner-White & Poet, 1986*) represented with a sample of sequences in a simplified notation (*h*-hydrophobic, *l*-loop-friendly, *x*-any), and with an idealized schematic structure. (B) Rules of a probabilistic context-free grammar modeling the subfamily. Set of terminal symbols of the grammar (the alphabet) consists of 20 amino acid identities. Lexical non-terminals *h, l* and *x* correspond to symbols of the simplified notation. They are mapped to terminal symbols (amino acids) through lexical rules (here, they have uniform probabilities for the sake of simplicity). Rules rewriting structural non-terminals *s* (the start symbol), *t* and *u* model the ladder of the hairpin, and the two-residue loop (*t → ll*). The grammar allows for bulges using non-terminal *b* and associated rules. (C) Fictitious query sequence (and its contact map) to be tested against the grammar. Possible mappings from amino acids to lexical non-terminals are shown below the sequence. Spatial proximity of residues is marked in the contact map with a circle. Empty circles denote trivial contacts between adjacent residues; filled circles denote spatial contacts between residues distant in the sequence. (D) Two possible derivations of the query sequence using the grammar. In each step, the left-most structural non-terminal is rewritten with a grammar rule. Final steps from lexical non-terminals to terminal symbols are combined for the sake of brevity. First derivation is apparently ca. 1,000 times more probable given the grammar. (E) Parse trees corresponding to the two derivations. Nodes representing terminal symbols and their incoming edges are omitted for the sake of clarity. If application of the rule *s → hth* is identified with generating hydrogen bonds between the two hydrophobic residues, the parse trees correspond to the two schematic structures. Note that only the left-hand-side tree captures all three distant contacts present in the contact map.

to solutions which do not significantly outperform HMMs (*Dyrka & Nebel, 2009*; *Dyrka, Nebel & Kotulska, 2013*). However, some studies hinted that CF level of expressiveness brought an added value in protein modeling when grammars fully benefiting from CF nesting and branching rules were compared in the same framework to grammars effectively limited to linear (regular) rules (*Dyrka, 2007*; *Dyrka, Nebel & Kotulska, 2013*). Good preliminary results were also obtained on learning sub-classes of CF grammars to model protein families, showing the interest of taking into account long-distance correlations in comparison to regular models (*Coste, Garet & Nicolas, 2012*; *Coste, Garet & Nicolas, 2014*). An important advantage of CF and CS grammars is that grammars themselves, and especially the syntactic analyses of the sequences according to the grammar rules, are human readable. For CF grammars, the syntactic analysis of one sequence can be represented by a parse tree showing one hierarchical application of grammar rules enabling to recognize the sequence (see Figs. 1B and 1E example). In RNA modeling, the shape of parse trees can be used for secondary structure prediction (*Dowell & Eddy, 2004*). In protein modeling, it was suggested that the shape of parse trees corresponded to protein spatial structures (*Dyrka & Nebel, 2009*), and that parse trees could convey biologically relevant information (*Sciacca et al., 2011*; *Dyrka, Nebel & Kotulska, 2013*).

## Grammar estimation with structural constraints

In this piece of research the focus is on learning probabilistic context-free grammars (PCFG) (*Booth, 1969*). This represents a trade-off between expressiveness of the model and computational complexity of the sequence recognition, which is cubic in time with regard to the input length.

Learning PCFG aims at shifting the probability mass from the entire space of possible sequences and their syntactic trees to the target population, typically represented by a sample. The problem is often confined to assigning probabilities to fixed production rules of a generic underlying non-probabilistic CFG (*Lari & Young, 1990*). Typically, the goal is to estimate the probabilistic parameters to get a grammar maximizing the likelihood of the (positive) sample, while, depending on the target application, other approaches also exist. For example, the contrastive estimation aims at obtaining grammars discriminating the target population from its neighborhood (*Smith & Eisner, 2005*).

The training sample can be made of a set of sequences or a set of syntactic trees. In the former case, all derivations for each sentence are considered valid. For a given underlying non-probabilistic CFG, probabilities of its rules can be estimated from sentences in the classical Expectation Maximization framework, e.g., the Inside-Outside algorithm (*Baker, 1979*; *Lari & Young, 1990*). However, the approach is not guaranteed to find the globally optimal solution (*Carroll & Charniak, 1992*). Heuristic methods applied for learning PCFG from positive sequences include also iterative biclustering of bigrams (*Tu & Honavar, 2008*), and genetic algorithms using a learnable set of rules (*Kammeyer & Belew, 1996*; *Keller & Lutz, 1998*; *Keller & Lutz, 2005*) or a fixed covering set of rules (*Tariman, 2004*; *Dyrka & Nebel, 2009*).

Much more information about the language is conveyed when syntactic trees, constraining the set of admissible parse trees, are given. (Throughout this paper the notion

of *parse tree* is reserved for syntactic trees generated by parsing with a specific grammar.) If available, a set of trees (a treebank) can be directly used to learn a PCFG (*Charniak, 1996*). Usability of information on the syntactic structure of sequences is highlighted by the result showing that a large class of non-probabilistic CFG can be learned from unlabeled syntactic trees (called also *skeletons*) of the training sample (*Sakakibara, 1992*). Algorithms for learning probabilistic CF languages, which exploit structural information from syntactic trees, have been proposed (*Sakakibara et al., 1993*; *Eddy & Durbin, 1994*; *Carrasco, Oncina & Calera-Rubio, 2001*; *Cohen et al., 2014*). An interesting middle way between plain sequences and syntactic trees are partially bracketed sequences, which constrain the shape of the syntactic trees (skeletons) but not node labels. The approach was demonstrated to be highly effective in learning natural languages (*Pereira & Schabes, 1992*). It was also applied to integrating uncertain information on pairing of nucleotides of RNA (*Knudsen, 2005*), by modifying the bottom-up parser to penalize probabilities of inconsistent derivations with respect to available information on nucleotide pairing and adjusting the amount of the penalty according to certainty of the structural information.

## Protein contact constraints

To our knowledge, constrained sets of syntactic trees have never been applied for estimating PCFG for proteins. In this research we propose to use spatial contacts between amino acids, possibly distant in the sequence, as a source of constraints. Indeed, an interaction forming dependency between amino acids usually requires a contact between them, defined as spatial proximity. Until recently, extensive contact maps were only available for proteins with experimentally solved structures, while individual interactions could be determined through mutation-based wet experiments.

Currently, reasonably reliable contact maps can also be obtained computationally from large collective alignments of evolutionary related sequences. The rationale for contact prediction is that if amino acids at a pair of positions in the alignment interact then a mutation at one position of the pair often requires a compensatory mutation at the other position in order to maintain the interaction intact. Since only proteins maintaining interactions vital for function successfully endured the natural selection, an observable correlation in amino acid variability at a pair of positions is expected to indicate interaction. However, standard correlations are transitive and therefore cannot be immediately used as interaction predictors. A break-through was achieved recently by Direct Coupling Analysis (DCA) (*Weigt et al., 2009*), which disentangles direct from indirect correlations by inferring a model on the alignment which can give information on the interaction strength of the pairs. There are different DCA methods based on how the model, which is usually a type of Markov Random Field, is obtained (*Morcos et al., 2011*; *Jones et al., 2012*; *Ekeberg et al., 2013*; *Kamisetty, Ovchinnikov & Baker, 2013*; *Seemayer, Gruber & Söding, 2014*; *Baldassi et al., 2014*). The state-of-the-art DCA-based meta-algorithms achieve mean precision in the range 42–74% for top $L$ predicted contacts and 69–98% for top $L/10$ predicted contacts, where $L$ is the protein length (*Wang et al., 2017*). Precision is usually lower for shorter sequences and especially for smaller alignments, however a few top hits may still provide relevant information (*Daskalov, Dyrka & Saupe, 2015*).

### Contributions of this research

In the broader plan, this research aims at developing a protein sequence analysis method advancing the current state of the art represented by the profile HMMs in being not limited to alignment-defined protein sequence families, and capable of capturing interactions between amino acids. The ideal approach would be based on the probabilistic (mildly) context-sensitive grammars, however their computational complexity significantly hampers practical solutions. Therefore, an intermediate approach based on the probabilistic context-free grammars is considered here, which is computationally cheaper and can represent the non-crossing (and non-overlapping) interactions between amino acids. Still, the main difficulty is efficient estimation of the grammars. Our solution is to accommodate information of protein contacts as syntactic structural constraints for the model estimation and, if possible, for the sequence analysis. The first contribution of this work consists on developing a theoretical framework for defining the maximum-likelihood and contrastive estimators of PCFG using contact constraints ('Estimation schemes using contact constraints'). Building on this general framework, the second contribution of this work is extension of our previous probabilistic context-free grammatical model for protein sequences (*Dyrka, 2007*; *Dyrka & Nebel, 2009*; *Dyrka, Nebel & Kotulska, 2013*), proposed in 'Application to contact grammars'. The extended model is evaluated with reference to the original one in the same evolutionary framework for inferring probabilities of grammar rules (*Dyrka & Nebel, 2009*), as described in 'Evaluation' (part of the 'Methods'). The assessment focuses on capability of acquiring contact constraints by the grammar (*descriptive performance*), and its effect on *discriminative performance* ('Results'). After the evaluation, an example using this method in a practical setting is presented. Finally, the potential of our approach beyond the current state of the art is demonstrated by creating a grammatical model of a meta-family of protein motifs. This piece of work finishes with discussion of the results ('Discussion'), followed by conclusions with analysis of limitations and perspectives for future work ('Conclusions').

## METHODS

We first show in 'Estimation schemes using contact constraints' how contact constraints can formally be introduced to get new generic maximum-likelihood and contrastive estimation schemes, and present then in 'Application to contact grammars' a practical implementation of these schemes on a simple generic form of grammars representing contacts.

### Estimation schemes using contact constraints

This section provides the mathematical basis for our method for training probabilistic context-free grammars (PCFG) from protein sequences annotated with pairwise contacts. Standard notations used in the field of grammar inference are introduced, complemented with a less common notion of the unlabeled syntactic tree which is the syntactic tree stripped from the syntactic variables ('Basic notations'). We propose to define the syntactic tree of a protein sequence as *consistent* with the contact map if for each pair of positions in contact, the path between corresponding leaves in the tree is shorter than given threshold (Eq. (1) in 'Contact constraints'). Finally, the maximum-likelihood and the contrastive

estimators formulæ are derived for training PCFG over the sets of unlabeled syntactic trees consistent with contact maps (Eqs. (2)–(4) in 'Estimation').

### Basic notations

Let $\Sigma$ be a non-empty finite set of atomic symbols (representing for instance amino acid species). The set of all finite strings over this alphabet is denoted by $\Sigma^*$. Let $|x|$ denote the length of a string $x$. The set of all strings of length $n$ is denoted by $\Sigma^n = \{x \in \Sigma^* : |x| = n\}$. Let $x = x_1 \ldots x_n$ be a sequence in $\Sigma^n$.

*Unlabeled syntactic tree.* An unlabeled syntactic tree (UST) $u$ for $x$ is an ordered rooted tree such that the leaf nodes are labeled by $x$, which is denoted as $yield(u) = x$, and the non-leaf nodes are unlabeled. Let $\mathcal{U}_*$ denotes the set of all USTs that yield a sequence in $\Sigma^*$, let $\mathcal{U}_n = \{u \in \mathcal{U}_* : yield(u) \in \Sigma^n\}$, where $n$ is a positive integer, and let $\mathcal{U}_x = \{u \in \mathcal{U}_* : yield(u) = x \in \Sigma^*\}$. Note that $\forall(x, w \in \Sigma^*, x \neq w) \; \mathcal{U}_x \cap \mathcal{U}_w = \varnothing$ and $\mathcal{U}_* = \cup_{x \in \Sigma^*} \mathcal{U}_x$. Moreover, let $U$ denotes an arbitrary subset of $\mathcal{U}_*$.

*Context-free grammar.* A context-free grammar (CFG) is a quadruple $G = \langle \Sigma, V, v_0, R \rangle$, where $\Sigma$ is defined as above, $V$ is a finite set of non-terminal symbols (also called variables) disjoint from $\Sigma$, $v_0 \in V$ is a special start symbol, and $R$ is a finite set of rules rewriting from variables into strings of variables and/or terminals $R = \{r_i : V \rightarrow (\Sigma \cup V)^*\}$ (see Fig. 1B). Let $\alpha = \alpha_1 \ldots \alpha_k$ be a sequence of symbols in $(\Sigma \cup V)^k$ for some natural $k$. A (left-most) derivation for $G$ is a string of rules $r = r_1 \ldots r_l \in R^l$, which defines an ordered parse tree $y$ starting from the root node labeled by $v_0$. In each step, by applying a rule $r_i : v_j \rightarrow \alpha_1 \ldots \alpha_k$, tree $y$ is extended by adding edges from the already existing left-most node labeled $v_j$ to newly added nodes labeled $\alpha_1$ to $\alpha_k$. Therefore, there is a one-to-one correspondence between derivation $r$ and parse tree $y$ (see Figs. 1D, 1E). Derivation $r$ is complete if all leaf nodes of the corresponding (complete) parse tree $y$ are labeled by symbols in $\Sigma$. Sets $\mathcal{Y}_*$, $\mathcal{Y}_n$ and $\mathcal{Y}_x$ denote parse tree sets generated with $G$ analogously as for the USTs. For a given parse tree $y$, $u(y)$ denotes the unlabeled syntactic tree obtained by removing the non-leaf labels on $y$. Given a UST $u$, let $\mathcal{Y}_G(u)$ be the set of all parse trees for grammar $G$ such that $u(y) = u$. For a set of USTs $U$, $\mathcal{Y}_G(U) = \cup_{u \in U} \mathcal{Y}_G(u)$. Note that $\forall(u, v \in U, u \neq v) \; \mathcal{Y}_G(u) \cap \mathcal{Y}_G(v) = \varnothing$.

*Probabilistic context-free grammar.* A probabilistic context-free grammar (PCFG) is a quintuple $\mathcal{G} = \langle \Sigma, V, v_0, R, \theta \rangle$, where $\theta$ is a finite set of probabilities of rules: $\theta = \{\theta_i = \theta(r_i) : R \rightarrow [0, 1]\}$, setting for each rule $v_k \rightarrow \alpha$ its probability to be chosen to rewrite $v_k$ with respect to other rules rewriting $v_k$ (such that $\forall(v_k \in V) \sum_{v_k \rightarrow \alpha} \theta(v_k \rightarrow \alpha) = 1$, see Fig. 1B). Let PCFG $\mathcal{G}$ that enhances the underlying non-probabilistic CFG $G = \langle \Sigma, V, v_0, R \rangle$ is denoted by $\mathcal{G} = \langle G, \theta \rangle$. The probability of parse tree $y$ using the probability measure induced by $\mathcal{G}$ is given by the probability of the corresponding derivation $r = r_1 \ldots r_l$:

$$prob(y|\mathcal{G}) = prob(r|\mathcal{G}) = \prod_{i=1}^{l} \theta(r_i).$$

$\mathcal{G}$ is said to be *consistent* when it defines probability distribution over $\mathcal{Y}_*$:

$$prob(\mathcal{Y}_*|\mathcal{G}) = \sum_{y \in \mathcal{Y}_*} prob(y|\mathcal{G}) = 1.$$

The probability of sequence $x \in \Sigma^*$ given $\mathcal{G}$ is:

$$prob(x|\mathcal{G}) = prob(\mathcal{Y}_x|\mathcal{G}) = \sum_{y \in \mathcal{Y}_x} prob(y|\mathcal{G}),$$

and the probability of UST $u \in \mathcal{U}_x$ given $\mathcal{G}$ is:

$$prob(u|\mathcal{G}) = prob(\mathcal{Y}_G(u)|\mathcal{G}) = \sum_{y \in \mathcal{Y}_G(u)} prob(y|\mathcal{G}).$$

Since $\mathcal{Y}_x$ and $\mathcal{Y}_G(u)$ define each a partition of $\mathcal{Y}_*$ for $x \in \Sigma^*$ and for $u \in \mathcal{U}_*$, a consistent grammar $\mathcal{G}$ defines also a probability distribution over $\Sigma^*$ and $\mathcal{U}_*$.

### Contact constraints

Most protein sequences fold into complex spatial structures. Two amino acids at positions $i$ and $j$ in the sequence $x$ are said to be in contact if distance between their coordinates in spatial structure $d(i,j)$ is below a given threshold $\tau$. A full contact map for a protein of length $n$ is a binary symmetric matrix $\mathsf{m}^{full} = (m_{i,j})_{n \times n}$ such that $m_{i,j} = [d(i,j) < \tau]$, where $[x]$ is the Iverson bracket (see Fig. 1C). Usually only a subset of the contacts is considered (see 'Protein contact constraints'). A (partial) contact map for a protein of length $n$ is a binary symmetric matrix $\mathsf{m} = (m_{i,j})_{n \times n}$ such that $m_{i,j} = 1 \implies d(i,j) < \tau$. Let $d_u(i,j)$ be the length of the shortest path from $i$th to $j$th leaf in UST $u$ for $x$. Given a threshold $\delta$, UST $u$ is said to be consistent with a contact map $\mathsf{m}$ of length $n$ if

$$m_{i,j} = 1 \implies d_u(i,j) < \delta. \tag{1}$$

For a contact map $\mathsf{m}$ of length $n$, let $\mathcal{U}_n^{\mathsf{m}}$ denotes the subset of $\mathcal{U}_n$ consistent with $\mathsf{m}$, and $\mathcal{U}_x^{\mathsf{m}}$ denotes the subset of $\mathcal{U}_x$ consistent with $\mathsf{m}$. Note that $\mathcal{U}_x^{\mathsf{m}} = \mathcal{U}_n^{\mathsf{m}} \cap \mathcal{U}_x$. Analogous notations apply to parse trees.

### Estimation

Learning grammar $\mathcal{G} = \langle \Sigma, V, v_0, R, \theta \rangle$ can be seen as inferring the unfixed components of $\mathcal{G}$ with the aim of shifting the probability mass from the entire space of unlabeled syntactic trees $\mathcal{U}_*$ to the set of unlabeled syntactic trees for the target population $\mathfrak{U}_{target}$. In practice, only a sample of the target population can be used for learning, hence estimation is performed on $\mathfrak{U}_{sample} \subseteq \mathfrak{U}_{target}$. Note that even in the most general case the set of terminal symbols $\Sigma$ is implicitly determined by the sample; moreover the start symbol $v_0$ is typically also fixed. A common special case considered in this work confines learning grammar $\mathcal{G}$ to estimating $\theta$ for a fixed quadruple of non-probabilistic parameters $\langle \Sigma, V, v_0, R \rangle$ (which fully determine the non-probabilistic grammar $G$ underlying $\mathcal{G}$). Given inferred grammar $\mathcal{G}_*$ and a query set of unlabeled syntactic trees $\mathcal{U}_{query}$, probability $prob(\mathcal{U}_{query}|\mathcal{G}_*)$ is an estimator of the likelihood that $\mathcal{U}_{query}$ belongs to population $\mathfrak{U}_{target}$.

*Maximum-likelihood grammar.* Let $X$ be a sample set of sequences in $\Sigma^*$, and let $\mathsf{M}$ be a set of corresponding contact matrices. The sample set $\mathcal{S} = [X\mathsf{M}]$ consists of a set of tuples $(x, \mathsf{m})$, where $x \in X$ and $\mathsf{m} \in \mathsf{M}$. Let $\mathfrak{U}_X^{\mathsf{M}}$ be the corresponding set of compatible USTs:

$$\mathfrak{U}_X^{\mathsf{M}} = \{\mathcal{U}_x^{\mathsf{m}} : (x, \mathsf{m}) \in \mathcal{S}\}.$$

Grammar $\mathcal{G}$ that concentrates probability mass on $\mathfrak{U}_X^{\mathsf{M}}$ can be estimated using the classical Bayesian approach:

$$\mathcal{G}_* = \arg\max_{\mathcal{G}} prob(\mathcal{G}|\mathfrak{U}_X^{\mathsf{M}}) = \arg\max_{\mathcal{G}} \frac{prob(\mathcal{G}) \cdot prob(\mathfrak{U}_X^{\mathsf{M}}|\mathcal{G})}{prob(\mathfrak{U}_X^{\mathsf{M}})}.$$

Noting that $prob(\mathfrak{U}_X^{\mathsf{M}})$ does not influence the result and, in the lack of prior knowledge, assuming $prob(\mathcal{G})$ uniformly distributed among all $\mathcal{G}$, the solution is then given by the maximum likelihood formula:

$$\mathcal{G}_* = \arg\max_{\mathcal{G}} prob(\mathcal{G}|\mathfrak{U}_X^{\mathsf{M}}) \simeq \mathcal{G}_{\mathrm{ML}} = \arg\max_{\mathcal{G}} prob(\mathfrak{U}_X^{\mathsf{M}}|\mathcal{G}).$$

Assuming independence of $\mathcal{U}_x^{\mathsf{m}}$s:

$$\mathcal{G}_{\mathrm{ML}} = \arg\max_{\mathcal{G}} \prod_{\mathcal{U}_x^{\mathsf{m}} \in \mathfrak{U}_X^{\mathsf{M}}} prob(\mathcal{U}_x^{\mathsf{m}}|\mathcal{G}) = \arg\max_{\mathcal{G}} \prod_{(x,\mathsf{m}) \in \mathcal{S}} \sum_{y \in \mathcal{Y}_x^{\mathsf{m}}} prob(y|\mathcal{G}). \tag{2}$$

In the absence of contact constraints, the maximization problem becomes equivalent to the standard problem of estimating grammar $\mathcal{G}$ given the sample $X$:

$$\mathcal{G}_{\mathrm{ML}}^{\mathsf{m}=0} = \arg\max_{\mathcal{G}} \prod_{\mathcal{U}_x \in \mathfrak{U}_X} prob(\mathcal{U}_x|\mathcal{G}) = \arg\max_{\mathcal{G}} \prod_{x \in X} \sum_{y \in \mathcal{Y}_x} prob(y|\mathcal{G}),$$

where $\mathsf{m} = 0$ denotes a square null matrix of size equal to the length of the corresponding sequence, and $\mathfrak{U}_X = \{\mathcal{U}_x^{\mathsf{m}=0} : x \in X\}$.

*Contrastive estimation.* Occasionally, it is reasonable to expect that $\mathcal{U}_{\mathrm{query}}$ comes from a neighborhood of the target population $\mathcal{N}(\mathfrak{U}_{\mathrm{target}}) \subset \mathcal{U}_*$. In such cases it is practical to perform *contrastive estimation* (*Smith & Eisner, 2005*), which aims at shifting the probability mass distributed by the grammar from the neighborhood of the of sample $\mathcal{N}(\mathfrak{U}_{\mathrm{sample}})$ to the sample itself $\mathfrak{U}_{\mathrm{sample}}$, such that:

$$\mathcal{G}_{\mathrm{CE}} = \arg\max_{\mathcal{G}} \prod_{\mathcal{U}_x \in \mathfrak{U}_{\mathrm{sample}}} \frac{prob(\mathcal{U}_x|\mathcal{G})}{prob(\mathcal{N}(\mathcal{U}_x)|\mathcal{G})}.$$

Consider two interesting neighborhoods. First, assume that contact map $\mathsf{m}$ is known and shared in the entire target population and hence in the sample: $\mathfrak{U}_X^{\mathsf{m}} = \{\mathcal{U}_x^{\mathsf{m}} : x \in X\}$. This implies the same length $n$ of all sequences. Then $\mathcal{U}_n^{\mathsf{m}}$ is a reasonable neighborhood of the target population, so

$$\mathcal{G}_{\mathrm{CE}(\mathsf{m})} = \arg\max_{\mathcal{G}} \prod_{\mathcal{U}_x^{\mathsf{m}} \in \mathfrak{U}_X^{\mathsf{m}}} \frac{prob(\mathcal{U}_x^{\mathsf{m}}|\mathcal{G})}{prob(\mathcal{U}_n^{\mathsf{m}}|\mathcal{G})} = \arg\max_{\mathcal{G}} \frac{\prod_{x \in X} \sum_{y \in \mathcal{Y}_x^{\mathsf{m}}} prob(y|\mathcal{G})}{\left[\sum_{y \in \mathcal{Y}_n^{\mathsf{m}}} prob(y|\mathcal{G})\right]^{|X|}}. \tag{3}$$
Second, assume that sequence $x$ is known to be yielded by the target population. Now, the goal is to maximize likelihood that the shapes of parse trees generated for sequences in the target population are consistent with contact maps. Then $\mathfrak{U}_X$ is a reasonable neighborhood of the sample $\mathfrak{U}_X^M$, so

$$\mathcal{G}_{\text{CE}(X)} = \arg\max_{\mathcal{G}} \prod_{(x,\mathsf{m})\in\mathcal{S}} \frac{prob(\mathcal{U}_x^{\mathsf{m}}|\mathcal{G})}{prob(\mathcal{U}_x|\mathcal{G})} = \arg\max_{\mathcal{G}} \prod_{(x,\mathsf{m})\in\mathcal{S}} \frac{\sum_{y\in\mathcal{Y}_x^{\mathsf{m}}} prob(y|\mathcal{G})}{\sum_{y\in\mathcal{Y}_x} prob(y|\mathcal{G})}. \tag{4}$$

## Application to contact grammars

We introduce here in 'Definitions' a simple form for context-free grammars, referred to as the Chomsky Form with Contacts (CFC), that supplements the classical Chomsky Normal Form (CNF) with *contact rules* to enable representing non-overlapping pairwise contacts between amino acids. The toy grammar in Fig. 1B provides an example CFC, with one contact rule $s \to hth$ generating a pair of amino acids in contact through lexical rules rewriting the $h$ symbols (e.g., $h \to V$, $h \to I$). The shortest path in the syntactic tree between such a pair of residues is then of length 4, the minimal path length between terminals for CFC grammars. We propose to use that threshold for defining the consistency of a syntactic tree with a contact map. This natural choice allows for computing Eqs. (2), (3) and (4) in polynomial (cubic) time with regard to the sequence length, as demonstrated in 'Parsing' and 'Calculating $prob(\mathcal{U}_n^{\mathsf{m}}|\ddot{\mathcal{G}})$'.

### Definitions

Let $\ddot{\mathcal{G}} = \langle \Sigma, V, v_0, R, \theta \rangle$ be a probabilistic context-free grammar such that $V = V_T \uplus V_N$, $R = R_a \uplus R_b \uplus R_c$, and

$R_a = \{r_i : V_T \to \Sigma\}$,
$R_b = \{r_j : V_N \to (V_N \cup V_T)(V_N \cup V_T)\}$,
$R_c = \{r_k : V_N \to V_T V_N V_T\}$.

Subsets $R_a$, $R_b$ and $R_c$ are referred to as *lexical*, *branching*, and *contact* rules, respectively. Joint subset $R_b \cup R_c$ is referred to as *structural* rules. Grammars which satisfy these conditions are hereby defined to be in the *Chomsky Form with Contacts* (CFC). It happens that the toy grammar in Fig. 1B is in CFC. When a CFC grammar satisfies $R_c = \varnothing$, it is in the Chomsky Normal Form (CNF).

Non-terminal symbols in $V_T$, which can be rewritten only into terminal symbols are referred to as *lexical* non-terminals, while non-terminal symbols in $V_N$ are referred to as *structural* non-terminals. Comparing the CFC grammar with the profile HMM, each match state of the latter can be identified with a unique lexical non-terminal, and emissions from a given state—with a set of lexical rules rewriting the non-terminal corresponding to the state.

Let $\mathsf{m}$ be a contact matrix compatible with the context-free grammar, i.e., no pair of positions in contact overlaps nor crosses boundaries of other pairs in contact (though pairs can be nested one in another):

$$\forall(i,j)\ m_{i,j} = 1 \wedge (i \le k \le j \oplus i \le l \le j) \Rightarrow m_{k,l} = 0,$$

where $\oplus$ denotes the exclusive disjunction, and positions in contact are separated from each other by at least 2:

$$\forall (i,j)\ i < j+2.$$

Let distance threshold in tree $\delta = 4$. Then a complete parse tree $y$ generated by $\ddot{\mathcal{G}}$ is consistent with m only if for all $m_{i,j} = 1$ derivation

$$\alpha_{1,i-1}\ v_k\ \alpha_{j+1,n} \overset{*}{\Rightarrow} \alpha_{1,i-1}\ x_i\ v_l\ x_j\ \alpha_{j+1,n}$$

is performed with a string of production rules

$$[v_k \rightarrow v_t v_l v_u][v_t \rightarrow x_i][v_t \rightarrow x_j],$$

where $\alpha_{i,j} \in (\Sigma \cup V)^{j-i+1}$, $v_k, v_l \in V_N$ and $v_t, v_u \in V_T$.

According to this definition, the left-hand (right-hand) side parse tree in Fig. 1E is consistent (*not* consistent) with the contact map in Fig. 1C.

### *Parsing*

Given an input sequence $x$ of length $n$ and a grammar in the CFC form $\ddot{\mathcal{G}}$, $prob(x|\ddot{\mathcal{G}}) \equiv prob(\mathcal{Y}_x|\ddot{\mathcal{G}}) = \sum_{y \in \mathcal{Y}_x} prob(y|\ddot{\mathcal{G}})$ can be calculated in $O(n^3)$ by a slightly modified probabilistic Cocke-Kasami-Younger bottom-up chart parser (*Cocke, 1969*; *Kasami, 1965*; *Younger, 1967*). Indeed, productions in $R_a \uplus R_b$ conforms to the Chomsky Normal Form (*Chomsky, 1959*), while it is easy to see that productions in $R_c$ requires only $O(n^2)$. The algorithm computes $prob(x|\ddot{\mathcal{G}}) = prob(\mathcal{Y}_x|\ddot{\mathcal{G}})$ in chart table P of dimensions $n \times n \times |V|$, which effectively sums up probabilities of all possible parse trees $\mathcal{Y}_x$. In the first step, probabilities of assigning lexical non-terminals $V_T$ for each terminal in the sequence $x$ are stored in the bottom matrix $P_1 = P[1,:,:]$. Then, the table P is iteratively filled upwards with probabilities $P[j,i,v] = prob(v \overset{*}{\Rightarrow} x_i...x_{i+j-1}|v \in V, \ddot{\mathcal{G}})$. Finally, $prob(\mathcal{Y}_x^m|\ddot{\mathcal{G}}) = P[n,1,v_0]$.

New extended version of the algorithm (Fig. 2) computes $prob(\mathcal{Y}_x^m|\ddot{\mathcal{G}})$, i.e., it considers only parse trees $\mathcal{Y}_x^m$ which are consistent with m. To this goal it uses an additional table C of dimensions $\sum(m)/2 \times n \times |V_T|$. After completing $P_1$ (lines 10–12), probabilities of assigning lexical non-terminals $V_T$ at positions involved in contacts are moved from $P_1$ to C (lines 13–21) such that each matrix $C_p = C[p,:,:]$ corresponds to $p$-th contact in m. In the subsequent steps C can only be used to complete productions in $R_c$; moreover both lexical non-terminals have to come either from $P_1$ or C, they can never be mixed (lines 35–40). The computational complexity of the extended algorithm is still $O(n^3)$ as processing of productions in $R_c$ has to be multiplied by iterating over the number of contact pairs in m, which is $O(n)$ since the cross-serial dependencies are not allowed.

### *Calculating $prob(\mathcal{U}_n^m|\ddot{\mathcal{G}})$*

This section shows effective computing $prob(\mathcal{U}_n^m|\ddot{\mathcal{G}})$, which is the denominator for the contrastive estimation of $\mathcal{G}_{CE(m)}$ (cf. 'Estimation'). Given a sequence $x$ of length $n$, a corresponding matrix $m$ of size $n \times n$ and a grammar $\ddot{\mathcal{G}}$, the probability of the set of trees over any sequence of length $n$ consistent with m is

$$prob(\mathcal{U}_n^m|\ddot{\mathcal{G}}) \equiv \sum_{x \in \Sigma^n} prob(\mathcal{U}_x^m|\ddot{\mathcal{G}}) = \sum_{x \in \Sigma^n} \sum_{y \in \mathcal{Y}_x^m} prob(y|\ddot{\mathcal{G}}).$$

```
01: function parse_cky_cm(x, m, Ra, Rb, Rc, Vt, Vn, v0)
02: # input:
03: # x - sequence, m - contact map
04: # Ra - lexical, Rb - branching, Rc - contact rules
05: # Vt - set of lexical, Vn - set of non-lexical non-terminals
06: # v0 - start symbol

07:     n = length(x)
08:     P[n, n, |Vn|+|Vt|] = 0.0
09:     C[sum(m)/2, n, |Vt|] = 0.0

10:     for i=1 to n
11:         for r in Ra
12:             if x[i]==r.rhs[1] P[1,i,r.lhs] = r.prob
13:     num_p=0
14:     for i=1 to n-2
15:         for j=i+2 to n
16:             if m[i,j]==1
17:                 for r in Ra
18:                     P[1,i,r.lhs] = P[1,j,r.lhs] = 0.0
19:                     if x[i]==r.rhs[1] C[p,i,r.lhs] = r.prob
20:                     if x[j]==r.rhs[1] C[p,j,r.lhs] = r.prob
21:                 num_p=num_p+1
22:     for j=2 to n
23:         for i=1 to n-j+1
24:             for k=1 to j-1
25:                 for r in Rb
26:                     P[j,i,r.lhs] += r.prob
27:                                     * P[ k,i,  r.rhs[1]]
28:                                     * P[j-k,i+k,r.rhs[2]]
29:             if (j>=3)
30:                 for r in Rc
31:                     P[j,i,r.lhs] += r.prob
32:                                     * P[1,  i,  r.rhs[1]]
33:                                     * P[j-2,i+1,r.rhs[2]]
34:                                     * P[1,  i+j,r.rhs[3]]
35:                 for c=0 to num_p-1
36:                     for r in Rc
37:                         P[j,i,r.lhs] += r.prob
38:                                         * C[p,  i,  r.rhs[1]]
39:                                         * P[j-2,i+1,r.rhs[2]]
40:                                         * C[p,  i+j,r.rhs[3]]
41:     return P[n, 1, v0]
```

**Figure 2  Pindocode of the modified CKY parser.**

Given grammar $\ddot{\mathcal{G}}$, any complete derivation $r$ is a composition $r = \dot{r} \circ \tilde{r}$, where $\dot{r} \in (R_a)^*$ and $\tilde{r} \in (R_b \cup R_c)^*$. Let $y$ be the parse tree corresponding to derivation $r$, and let $\tilde{y}$ be an incomplete parse tree corresponding to derivation $\tilde{r}$. Note that for any $y$ corresponding to $r = \dot{r} \circ \tilde{r}$ there exists one and only one $\tilde{y}$ corresponding to $\tilde{r}$. Let $\tilde{\mathcal{Y}}_x^{\mathrm{m}}$ denote the set of such incomplete trees $\tilde{y}$. Note that labels of the leaf nodes of $\tilde{y}$ are lexical non-terminals $\forall(i)\ \alpha_{i,i} \in V_T$, and that $\dot{r}$ represents the unique left-most derivation $yield(\tilde{y}) \overset{*}{\Rightarrow} x$. Thus,

$$\sum_{x \in \Sigma^n} \sum_{y \in \mathcal{Y}_x^{\mathrm{m}}} prob(y|\ddot{\mathcal{G}}) = \sum_{x \in \Sigma^n} \sum_{\tilde{y} \in \tilde{\mathcal{Y}}_x^{\mathrm{m}}} prob(\tilde{y}|\ddot{\mathcal{G}}) \cdot prob(yield(\tilde{y}) \overset{*}{\Rightarrow} x|\ddot{\mathcal{G}}).$$

Note that value of the expression will not change if the second summation is over $\tilde{y} \in \tilde{\mathcal{Y}}_n^{\mathrm{m}}$ since $\forall(\tilde{y} \notin \tilde{\mathcal{Y}}_x^{\mathrm{m}})\ prob(yield(\tilde{y}) \overset{*}{\Rightarrow} x|\ddot{\mathcal{G}}) = 0$. Combining with observation that $prob(\tilde{y}|\ddot{\mathcal{G}})$ does not depend on $x$, the expression can be therefore rewritten as:

$$\sum_{x \in \Sigma^n} \sum_{y \in \mathcal{Y}_x^{\mathrm{m}}} prob(y|\ddot{\mathcal{G}}) = \sum_{\tilde{y} \in \tilde{\mathcal{Y}}_n^{\mathrm{m}}} prob(\tilde{y}|\ddot{\mathcal{G}}) \cdot \sum_{x \in \Sigma^n} prob(yield(\tilde{y}) \overset{*}{\Rightarrow} x|\ddot{\mathcal{G}}).$$

However, if $\ddot{\mathcal{G}}$ is *proper*, then $\forall(\tilde{y} \in \tilde{\mathcal{Y}}_n^{\mathrm{m}})\ \sum_{x \in \Sigma^n} prob(yield(\tilde{y}) \overset{*}{\Rightarrow} x|\ddot{\mathcal{G}}) = 1$, as:

$$\sum_{x \in \Sigma^n} prob(yield(\tilde{y}) \overset{*}{\Rightarrow} x|\ddot{\mathcal{G}}) = \sum_{x \in \Sigma^n} \prod_{i=1}^n \theta(\alpha_{i,i} \to x_i) =$$
$$\sum_{x \in \Sigma^n} \theta(\alpha_{1,1} \to x_1) \cdot \ldots \cdot \theta(\alpha_{n,n} \to x_n) =$$
$$\theta(\alpha_{1,1} \to a_1) \cdot \theta(\alpha_{2,2} \to a_1) \cdot \ldots \cdot \theta(\alpha_{n-1,n-1} \to a_1) \cdot \theta(\alpha_{n,n} \to a_1) +$$
$$\theta(\alpha_{1,1} \to a_1) \cdot \theta(\alpha_{2,2} \to a_1) \cdot \ldots \cdot \theta(\alpha_{n-1,n-1} \to a_1) \cdot \theta(\alpha_{n,n} \to a_2) +$$
$$\vdots$$
$$\theta(\alpha_{1,1} \to a_{|\Sigma|}) \cdot \theta(\alpha_{2,2} \to a_{|\Sigma|}) \cdot \ldots \cdot \theta(\alpha_{n-1,n-1} \to a_{|\Sigma|}) \cdot \theta(\alpha_{n,n} \to a_{|\Sigma|}) =$$

$$\begin{pmatrix} \theta(\alpha_{1,1} \to a_1) \cdot \theta(\alpha_{2,2} \to a_1) \cdot \ldots \cdot \theta(\alpha_{n-1,n-1} \to a_1) + \\ \theta(\alpha_{1,1} \to a_1) \cdot \theta(\alpha_{2,2} \to a_1) \cdot \ldots \cdot \theta(\alpha_{n-1,n-1} \to a_2) + \\ \vdots \\ \theta(\alpha_{1,1} \to a_{|\Sigma|}) \cdot \theta(\alpha_{2,2} \to a_{|\Sigma|}) \cdot \ldots \cdot \theta(\alpha_{n-1,n-1} \to a_{|\Sigma|}) \end{pmatrix} \cdot \sum_{s=1}^{|\Sigma|} \theta(\alpha_{n,n} \to a_s),$$

where $a_s \in \Sigma$. Since $\ddot{\mathcal{G}}$ is *proper* then $\forall(v \in V_T)\ \sum_{s=1}^{|\Sigma|} \theta(v \to a_s) = 1$ and therefore the entire formula evaluates to 1, which can be easily shown by iterative regrouping. This leads to the final formula:

$$prob(\mathcal{U}_n^{\mathrm{m}}|\ddot{\mathcal{G}}) = \sum_{\tilde{y} \in \tilde{\mathcal{Y}}_n^{\mathrm{m}}} prob(\tilde{y}|\ddot{\mathcal{G}}).$$

Technically, $\sum_{\tilde{y} \in \tilde{\mathcal{Y}}_n^{\mathrm{m}}} prob(\tilde{y}|\ddot{\mathcal{G}})$ can be readily calculated by the bottom-up chart parser by setting $\forall(r_k \in R_a)\ \theta(r_k) = 1$.

## Evaluation

The present approach for learning PCFGs with the contact constraints was evaluated using our evolutionary framework for learning the probabilities of rules (*Dyrka & Nebel, 2009*;

*Dyrka, Nebel & Kotulska, 2013*). The underlying non-probabilistic CFGs were based on grammars used in our previous research (*Dyrka & Nebel, 2009*), which conformed to the Chomsky Normal Form (CNF) and consisted of an alphabet of twenty terminal symbols representing amino acid species

$$\Sigma = \{A, C, D, E, F, G, H, I, K, L, M, N, Q, P, R, S, T, V, W, Y\},$$

a set of non-terminals symbols $V = V_T \uplus V_N$, where $V_T = \{l_1, l_2, l_3\}$ and $V_N = \{v_0, v_1, v_2, v_3\}$, and a set of rules $R = R_a \uplus R_b$, which consisted of all possible allowed combinations of symbols, hence $|R_a| = 60, |R_b| = 196$. In addition, extended grammars $\ddot{G}$ in the Chomsky Form with Contacts (CFC) were constructed with added contact rules, $R = R_a \uplus R_b \uplus R_c$, again with all combinations of symbols ( $|R_c| = 144$). For the sake of transparent evaluation, combinations of symbols in the rules were not constrained beyond general definition of the CNF or CFC model, respectively, to avoid interference with the contact constraints. The number of non-terminal symbols was limited to a few in order to keep the number of parameters to be optimized by the genetic algorithm reasonably small. The small number of non-terminals implied relatively high generality of the resulting model, for example, only three distinct emission profiles of amino acids were defined by the lexical rules. The number of three lexical non-terminals was assumed from our previous research (*Dyrka & Nebel, 2009*; *Dyrka, Nebel & Kotulska, 2013*), in which lexical rule probabilities were fixed according to representative physicochemical properties of amino acids. In that setting, it seemed justified to have distinct symbols for the low, medium and high levels of the properties. Clearly, this has to be expected to confine specificity and limit attainable discriminatory power of the grammars. Although adjusting proportion of lexical and structural non-terminals could potentially improve performance of the grammatical model, it was not explored here, since the focus of evaluation was on the added value of the contact constraints for learning rule probabilities, rather than on the optimal set of rules.

### Learning

Our evolutionary learning framework used the genetic algorithm where each individual represented a whole grammar, the approach known as the Pittsburgh style (*Smith, 1980*). For a given underlying non-probabilistic CFG $\ddot{G}$ and the positive training sample, the framework estimated probabilities $\theta$ of the corresponding PCFG $\ddot{\mathcal{G}} = \langle \ddot{G}, \theta \rangle$. Unlike previous applications of the framework in which probabilities of the lexical rules were fixed according to representative physicochemical properties of amino acids (*Dyrka & Nebel, 2009*; *Dyrka, Nebel & Kotulska, 2013*), in this research probabilities of all rules were subject to evolution. The objective functions were implemented for the maximum-likelihood estimator $\ddot{\mathcal{G}}_{\text{ML}}$, and for the constrastive estimators $\ddot{\mathcal{G}}_{\text{CE}(X)}$ and $\ddot{\mathcal{G}}_{\text{CE}(m)}$. Besides, the setup of the genetic algorithm closely followed that of *Dyrka & Nebel (2009)*.

### Performance measures

Performance of grammars was evaluated using a variant of the 8-fold Cross-Validation scheme in which 6 parts are used for training, 1 part is used for validation and parameter selection, and 1 part is used for final testing and reporting results (the total of 56 combinations). The negative set was not used in the training phase. For testing, protein

sequences were scored against the null model (a unigram), which assumed global average frequencies of amino acids, no contact information, and the length of query sequence. The amino acid frequencies were obtained using the online ProtScale tool for the UniProtKB/Swiss-Prot database (*Gasteiger et al., 2005*).

*Discriminative performance.* Grammars were assessed on the basis of the average precision (AP) in the recall-precision curve (RPC). The advantage of RPC over the more common Receiver Operating Characteristic (ROC) is robustness to unbalanced samples where negative data is much more numerous than positive data (*Davis & Goadrich, 2006*). AP approximates the area under RPC.

*Descriptive performance.* Intuitively, a decent explanatory grammar generates parse trees consistent with the spatial structure of the analyzed protein. Therefore, the descriptive performance of grammar can be quantified as the amount of contact information encoded in the grammar and imposed on its derivations. In other words, it is expected that the grammar ensures that residues in contact are close in the parse tree (*Pyzik, Coste & Dyrka, 2019*). The most straightforward approach to measure the descriptive performance is to use the skeleton of the most likely parse tree as a predictor of spatial contacts between positions in a given protein sequence, parameterized by the cutoff $\delta$ on path length between the leaves. The natural threshold for grammar in the CFC form is $\delta = 4$ meaning that the pair of residues is predicted to be in contact if they are parsed with a contact rule. The precision at this threshold was reported for CFC grammars since the precision is the usual measure of contact prediction performance (*Wang et al., 2017*). In addition, AP of the RPC, which sums up over all possible cutoffs, was computed to allow comparison with grammars without pairing rules. Our recent research suggests that the measure is suitable for the contact-map-based comparison of the overall topology of parse trees generated with various grammars (*Pyzik, Coste & Dyrka, 2019*). Since our definition of consistency between the parse tree and the contact map imposes that inferred grammars maximize the recall rather than the precision of contact prediction, the learning process was assessed using the recall measured with regard to the partial contact map used in the training for $\delta = 4$. Local variants of the measures of descriptive performance can be defined to focus only on residues that are in contact with $k$-th residue. This can be obtained by using only respective row of the contact map $m_{k,\bullet}$ when calculating the value of a measure for the residue at position $k$. The local measures of descriptive performance can be used to assess the location of a residue in the parse tree (*Pyzik, Coste & Dyrka, 2019*).

*Implementation.* The PCFG-CM parser and the Protein Grammar Evolution framework were implemented in C++ using GAlib (*Wall, 2005*) and Eigen (*Guennebaud & Jacob, 2010*). Performance measures were implemented in Python 2 (*Van Rossum & De Boer, 1991*) using Biopython (*Cock et al., 2009*), igraph (*Csardi & Nepusz, 2006*), NumPy (*Van der Walt, Colbert & Varoquaux, 2011*), pyparsing (*McGuire, 2008*), scikit-learn (*Pedregosa et al., 2011*) and SciPy (*Jones, Oliphant & Peterson, 2001*).

Source code of PCFG-CM is available at https://git.e-science.pl/wdyrka/pcfg-cm under the GPL 3 license.

**Table 1** **Datasets.** *sim*—maximum sequence similarity, *npos/nneg*—number of positive/negative sequences, *len*—sequence length in amino acids, *ncon*—total number of non-local contacts (sequence separation 3 +), *msiz*—number of contacts selected for training.

| id | Type | Sim | npos | nneg | len | pdb | ncon | msiz |
|---|---|---|---|---|---|---|---|---|
| CaMn | binding-site | 71% | 24 | 28,560 | 27 | 2zbj | 41 | 6 |
| NAP | binding-site | 70% | 64 | 47,736 | 16 | 1mrq | 11 | 2 |
| HET-s | amyloid | 70% | 160 | 33,248 | 21 | 2kj3 | 10 | 3 |

# RESULTS

## Basic evaluation

### Materials

Probabilistic grammars were estimated for three samples of protein fragments related to functionally relevant gapless motifs (*Sigrist et al., 2002*; *Bailey & Elkan, 1994*). Within each sample, all sequences shared the same length, which avoided sequence length effects on grammar scores (this could be resolved by an appropriate null model). For each sample, one experimentally solved spatial structure in the Protein Data Bank (PDB) (*Berman et al., 2000*) was selected as a representative. The three samples included amino acid sequences of two small ligand binding sites (already analyzed in *Dyrka & Nebel (2009)* and a functional amyloid (Table 1):

- *CaMn*: a Calcium and Manganese binding site found in the legume lectins (*Sharon & Lis, 1990*). Sequences were collected according to the PROSITE PS00307 pattern (*Sigrist et al., 2013*) true positive and false negative hits. Original boundaries of the pattern were extended to cover the entire binding site, similarly to *Dyrka & Nebel (2009)*. The motif folds into a stem-like structure with multiple contacts, many of them forming nested dependencies, which stabilize anti-parallel beta-sheet made of two ends of the motif (Fig. 3A based on pdb:2zbj (*De Oliveira et al., 2008*));
- *NAP*: the Nicotinamide Adenine dinucleotide Phosphate binding site fragment found in an aldo/keto reductase family (*Bohren et al., 1989*). Sequences were collected according to the PS00063 pattern true positive and false negative hits (four least consistent sequences were excluded). The motif is only a part of the binding site of the relatively large ligand. Intra-motif contacts seem to be insufficient for defining the fold, which depends also on interactions with amino acids outside the motif (Fig. 3B based on pdb:1mrq (*Couture et al., 2003*));
- *HET-s*: the HET-s-related motifs r1 and r2 involved in the prion-like signal transduction in fungi identified in a recent study (*Daskalov, Dyrka & Saupe, 2015*). The largest subset of motif sequences with length of 21 amino acids was used to avoid length effects on grammar scores. When interacting with a related motif r0 from a cooperating protein, motifs r1 and r2 adopt the beta-hairpin-like folds which stack together. While stacking of multiple motifs from several proteins is essential for stability of the structure, interactions between hydrophobic amino acids within a single hairpin are also important. In addition, correlation analysis revealed strong dependency between positions 17 and

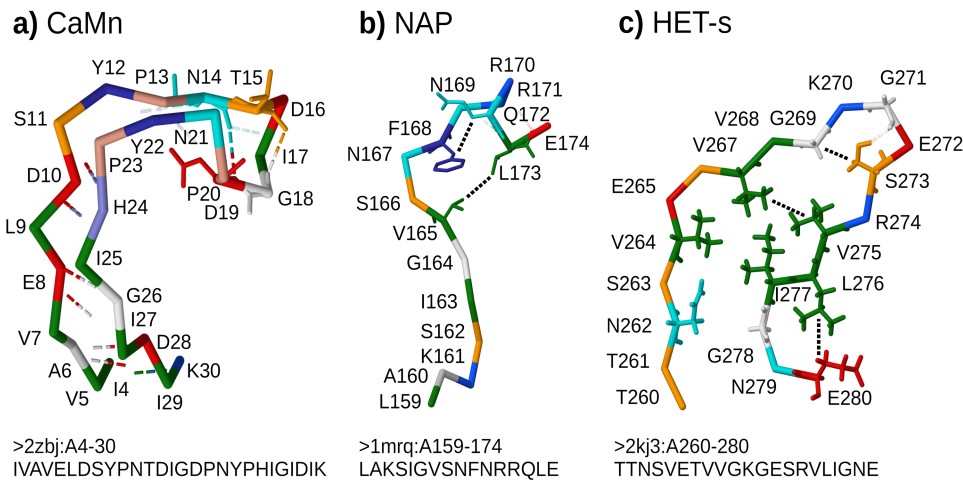

**Figure 3** **Representative structures of the sample motifs.** (A) Legume lectin Calcium and Manganese binding site; (B) Aldo/keto reductase NAP binding site fragment; (C) HET-s prion motif r2. Backbones are plotted with J(S)mol using the "amino" color scheme (*Herraez, 2006*; *Hanson et al., 2013*). Calculated hydrogen bonds are shown with dashed lines colored according to the interaction partners. Hydrogen bonds *not used* for defining contact maps are dimmed. Other contacts *used* for defining contact maps are shown with black dotted lines. Some side chains are shown for better visibility of selected bonds and contacts. For each structure, only a subset of interactions was chosen for defining the context-free-compatible *partial* contact map based on spatial proximity, hydrogen bonds (CaMn), and mutual correlation (HET-s). For example the pair of V264 and I277 in the HET-s structure conforms to definition of contact, however it was omitted since it crosses another contact between L276 and E280.

21 (*Daskalov, Dyrka & Saupe, 2015*) (corresponding to L276 and E280 in Fig. 3C based on *Van Melckebeke et al. (2010)*).

Negative samples were designed to roughly approximate the entire space of protein sequences. They were based on the negative set from (*Dyrka & Nebel, 2009*), which consisted of 829 single chain sequences of 300–500 residues retrieved from the Protein Data Bank (*Berman et al., 2000*) at identity of 30% (accessed on 12th December 2006). For each positive sample, the corresponding negative sample was obtained by cutting the basic negative set into overlapping subsequences of the length of positive sequences.

All samples were made non-redundant at level of sequence similarity around 70% using cd-hit (*Li & Godzik, 2006*), which significantly reduced their cardinalities. The threshold balanced the size of positive samples, distribution of their variability, and inter-fold diversity. Overall diversity of samples ranged from the most homogeneous CaMn (average identity of 49%) to the most diverse HET-s, which consisted of 5 subfamilies (*Daskalov, Dyrka & Saupe, 2015*) (average identity of 21%). The ratio between negative and positive samples was high and varied from 1190:1 for CaMn to 207:1 for HET-s. Contact pairings were assigned manually and collectively to all sequences in each set based on a selected representative spatial structure in the PDB database (Fig. 3).

**Table 2  Discriminative performance of grammars in terms of AP.**

| Grammar | CNF | CFC | | CFC | | CFC | |
| --- | --- | --- | --- | --- | --- | --- | --- |
| Estimation | ML | ML | | ML | | CE(m) | |
| Train w/contacts | n/a | no | | yes | | yes | |
| Test w/contacts | no | no | yes | no | yes | no | yes |
| CaMn | 0.94 | 0.96 | 0.67 | 0.95 | 0.95 | 0.79 | 0.98 |
| NAP | 0.78 | 0.86 | 0.28 | 0.75 | 0.79 | 0.24 | 0.91 |
| HET-s | 0.46 | 0.43 | 0.24 | 0.60 | 0.81 | 0.23 | 0.94 |

### Performance

The implementation of the framework for learning PCFGs for protein sequences using contact constraints, presented in 'Application to contact grammars' and 'Evaluation', is evaluated with reference to learning without the constraints. For grammars with the contact rules (CFC), probabilities of rules $\theta$ were estimated either using training samples made of sequences coupled with a contact map, or using sequences alone. For grammars without the contact rules (CNF), probabilities of rules were estimated using sequences alone, since these grammars cannot generate parse trees consistent with contact maps for the distance threshold $\delta = 4$.

*Discriminative power.* For evaluation of the discriminative power of the PCFG-CM approach, the rule probabilities were estimated using the maximum-likelihood estimator (denoted ML) and the contrastive estimator with regard to a given contact map (denoted CE(m)). The discriminative performance of the resulting probabilistic grammars for test data made of sequences alone and sequences coupled with a contact map is presented in Table 2 in terms of the average precision (AP).

The baseline is the average precision of CNF and CFC grammars estimated without contact constraints tested on sequences alone, which ranged from 0.43–0.46 for HET-s to 0.94–0.96 for CaMn. The scores show negative correlation with diversity of the samples and limited effect of adding contact rules (though the latter may result from more difficult learning of increased number of parameters with added rules). Grammars with the contact rules estimated without a contact map performed much worse when tested on the samples coupled with a contact map. This indicated that, in general, parses consistent with the constraints were not preferred by default when grammars were trained on sequences alone.

For all three samples, not surprisingly, the highest AP (0.91–0.98) achieved grammars obtained using the contrastive estimation with regard to a contact map tested on the samples with the same map. The improvement relative to the baseline was most pronounced for HET-s, yet still statistically significant ($p < 0.05$) for NAP. As expected, the contrastively estimated grammars performed poorly on sequences alone except for the CaMn sample.

The maximum-likelihood grammars estimated with a contact map and tested on sequences coupled with the same map performed worse than the contrastively estimated grammars but comparably or significantly better (HET-s) than the baseline. The average precision of these grammars was consistently lower when tested on sequences alone, yet still considerable (from 0.60 for HET-s to 0.95 for CaMn). It is notable that in the HET-s

**Table 3 Descriptive quality of the most likely parse trees derived from sequences alone.** In terms of recall at the distance threshold $\delta = 4$ w.r.t. the training contact map $m$, and precision at $\delta = 4$ (and AP over thresholds $\delta$) w.r.t. the full contact map of the reference *pdb* structure for sequence separation 3 +. Note that the shortest length of any path between leaves in the most likely parse trees of the CNF grammar equals 5, which makes measures using $\delta = 4$ unutile.

| Grammar | CNF | | CFC | | CFC | | CFC |
|---|---|---|---|---|---|---|---|
| Estimation | ML | | ML | | ML | | CE(X) |
| Train w/contacts | n/a | | no | | yes | | yes |
| Reference | pdb | m | pdb | m | pdb | m | pdb |
| CaMn | (0.24) | 0.45 | 0.69 (0.53) | 0.92 | 0.87 (0.66) | 0.98 | 0.84 (0.66) |
| NAP | (0.16) | 0.00 | 0.14 (0.12) | 0.96 | 0.64 (0.29) | 0.96 | 0.64 (0.29) |
| HET-s | (0.08) | 0.02 | 0.13 (0.14) | 0.79 | 0.52 (0.24) | 0.97 | 0.57 (0.27) |

case, the maximum-likelihood grammars estimated with a contact map achieved better AP on sequences alone than the maximum-likelihood grammars estimated without a contact map.

Universally high AP for CaMn can be contributed to the relatively strong pairing signal from the long stem-like part of the motif particularly suitable for modeling with the contact rules.

*Descriptive power.* For evaluation of the descriptive power of the PCFG-CM approach, the rule probabilities were estimated using the maximum-likelihood estimator (denoted ML) and the contrastive estimator with regard to the sequence set (denoted CE(X)). Descriptive value of the most probable parse trees generated using the resulting probabilistic grammars for test sequences without contact information is presented in Table 3. Efficiency of the learning was measured on the basis of the recall at the distance threshold $\delta = 4$ with regard to the context-free compatible contact map $m$ used in the training. Consistency of the most likely parse tree with the protein structure was measured on the basis of the precision of contact prediction at the distance threshold $\delta = 4$ with regard to all contacts in the reference spatial structure with separation in sequence of at least 3. Both measures are not suitable for assessing grammars without contact rules. Therefore, average precision over all thresholds $\delta$ was used as a complementary measure of consistency of the most likely trees with the protein structure. Note that the AP scores achievable for a context-free parse tree are reduced by overlapping of pairings.

The baseline is the result for grammars with the contact rules estimated without contact constraints. The most likely parse trees generated using these grammars conveyed practically no information about contacts for NAP and HET-s (recall w.r.t. contact map $m$ close to zero) and limited information about contacts for CaMn (moderate recall of 0.45), see Fig. 4. Learning with the contact constraints resulted in increase of the recall to 0.79–0.98, which testified efficiency of the process.

Importantly, consistency of the most likely parse trees with the protein structure measured by the precision followed a similar pattern and increased from 0.13 for HET-s, 0.14 for NAP, and 0.69 for CaMn when grammars with the contact rules were estimated without a contact map, to 0.52–0.57, 0.64, and 0.84–0.87, respectively, when grammars

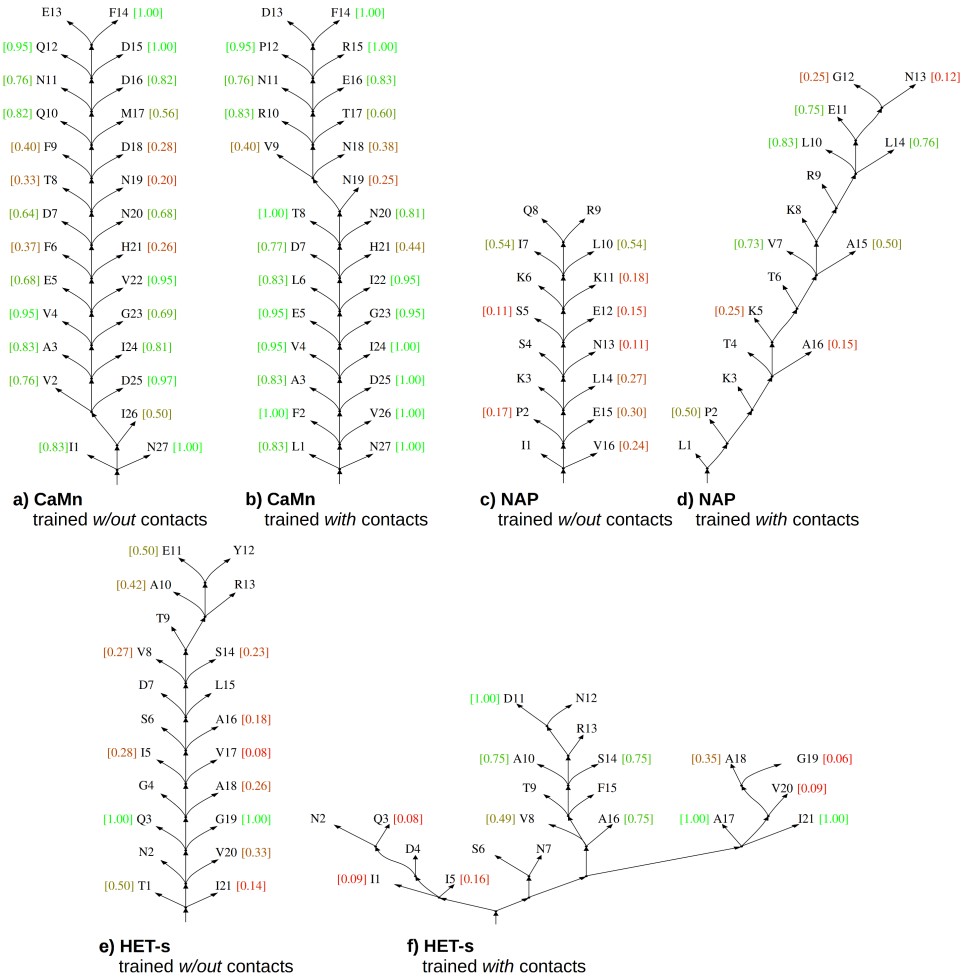

**Figure 4** Skeletons of most likely parse trees for selected positive test sequences obtained using grammars in the CFC form trained without and with the contact constraints. (A) CaMn tree according to grammar trained without contacts; (B) CaMn tree according to grammar trained with contacts; (C) NAP tree according to grammar trained without contacts; (D) NAP tree according to grammar trained with contacts; (E) HET-s tree according to grammar trained without contacts; (F) HET-s tree according to grammar trained with contacts. For each case, the tree of the *median* AP over all test runs and sequences is shown. Contact maps were *not used* for testing. Nodes corresponding to lexical non-terminal symbols are merged with terminal nodes (leaves of the trees) for the sake of simplicity. Terminal nodes are annotated with *local* AP calculated for each position (from 0.0 (bad, red) to 1.0 (perfect, green)). The minimum sequence separation of residues in contact of three or more is assumed; leaves with no intra-motif contacts outside this range are not scored.

were estimated with a contact map. Accordingly, evaluation in terms of the average precision over distance thresholds indicated that distances in the most likely parse trees better reflected the protein structure if grammars were trained with the contact constraints, as illustrated in Fig. 4.

## Sample applications
### Searching for related motifs

In this section probabilistic grammars for HET-s r1 and r2 motifs, learned in the proposed estimation scheme, are applied to solving a practical problem of searching for related r0 motifs in a limited-size dataset (around 1,000–5,000 sequences) based on (*Dyrka et al., 2014*; *Daskalov, Dyrka & Saupe, 2015*).

*Materials.* HET-s motifs r1 and r2 adopt the beta-hairpin-like fold when templated with the related motif r0 in the N-terminus of a cooperating NLR protein (*Seuring et al., 2012*). While the r0 motifs share a considerable sequence similarity with the interacting r1 and r2 motifs (average identity of around 30%), they contain significantly less aspartic acid, glutamic acid and lysine, and more histidine and serine (*Daskalov, Dyrka & Saupe, 2015*). A set of 98 HET-s r0 motifs was previously manually extracted from genes of NLR proteins adjacent to genes encoding proteins containing the r1 and r2 motifs (*Daskalov, Dyrka & Saupe, 2015*). Its subset of 77 non-redundant 21-residue long r0 motifs is later referred here as HET-s/r0. It can be reasonably expected that the r0 motifs can be automatically extracted from NLR proteins using grammars learned for the r1 and r2 motifs. As a proxy of this practical scenario, performance of discriminating the HET-s/r0 motifs against a set of 849 full-length NLR proteins with N-terminal known to contain a non-prion forming domain (*Dyrka et al., 2014*) was evaluated. (According to the current understanding of NLRs, it is highly unlikely that their N-terminal domain contains both a (possibly unnoticed) prion-forming motif and domain of other type (*Daskalov et al., 2015*).) In addition, the entire set of known 5765 fungal NLRs (*Dyrka et al., 2014*) was scanned for HET-s r0 motifs using the HET-s grammars. The results were compared with hits obtained using a profile HMM trained on the same data as the HET-s grammars, and the inhouse HET-s profile HMM from *Dyrka et al. (2014)*. Several variants of sets of grammar rules were investigated. Moreover, an alternative contact map with the pairing of positions 5 and 18 instead of 17 and 21 was tested (see Fig. 3). Each setup was run six times to account for expected randomness in the learning process.

*Evaluation.* The best fitting to the training sample was achieved with grammars which consisted of three lexical non-terminals, the start structural non-terminal rewritable into the branching and contact rules, two structural non-terminals rewritable into the branching rules, and four structural non-terminals rewritable into the contact rules (total of 10 non-terminals and 675 rules), and were estimated to optimize the maximum-likelihood using the alternative contact map. Importantly, learning with the alternative contact map substantially improved fitness to the training data in comparison to learning without any contact constraints (probability mass over the training set increased roughly 300 times on average over six runs).

The single best grammar achieved the average precision of 0.74 when used for discriminating HET-s/r0 motif from non-prionic NLR sequences (parsing without the contact map). The performance improved to AP of 0.82 when the mean score from six grammars was used for classifying. For the arbitrary threshold of 4 (or 5) of the mean log

probability ratio between the grammars and the null model (meaning that a given sequence is 10,000 (resp. 100,000) times more probable with the HET-s grammars than with the null), the precision was 0.59 (1.00) and the recall was 0.77 (0.58). While these scores are acceptable, especially taking into account simplicity of the grammars, they were below AP of 0.92 achieved with the profile HMM estimated on the same data using hmmer 3.1b2 with the standard parameters of training (*Eddy, 2011*). Yet, the recall for 100% precision was similar as for the grammars (0.79 at the bit score of 9.7). Scoring with the profile HMM was performed with the *–max* flag and effectively no *E*-value threshold, and separately for each overlapping 21-amino acid long fragment of the negative set.

Next, the six grammars were used for scanning the set of full-length fungal NLR sequences. With the threshold of the mean log probability ratio of 5, matches were found in 33 sequences. Out of them, 29 matches started within first twenty residues of relatively short N-terminal domains (up to 116 amino acids), as expected for the prion-forming domain. This included 18 HET-s r0 motifs from *Daskalov, Dyrka & Saupe (2015)*. Among the remaining 11 sequences with candidate r0 motifs, the corresponding r1 and r2 patterns were identified in adjacent genes in 6 cases (with the HET-s grammars or manually). The set of 33 sequences extracted with the grammars included 14 out of 15 HET-s annotations assigned with the inhouse profile HMM in *Dyrka et al. (2014)*.

### Making generalized descriptors

In this section the generalizing potential of PCFG descriptors is illustrated by learning a single grammar for two non-homologous but functionally related Calcium-binding motifs.

*Materials.* Calcium-binding sites, which are widely spread across many functional families of proteins, are formed by multiple various structural folds (*Bindreither & Lackner, 2009*). Two prominent families are the lectin legume beta-loop-beta motif (already described in 'Materials' under designation CaMn) and the EF hand alpha-loop-alpha motif (*Kawasaki & Kretsinger, 1995*). While apparently different, they are both continuous and involve the central loop (yet very different) participating in coordination of the Calcium ion (*Bindreither & Lackner, 2009*). These features made them an appealing target for investigating capability of the current grammatical framework for generalizing beyond a single family of sequences.

Our training set consisted of the entire CaMn sample (24 sequences), and the subset of EF hand motifs extracted—on the basis of the contact pattern—from the Calcium binding proteins of known spatial structure prepared for training the FEATURE model (*Zhou, Tang & Altman, 2015*). Boundaries of the EF hand motifs were specified to include the residues coordinating the Calcium ion, according to Ligplot (*Wallace, Laskowski & Thornton, 1995*), plus the envelope of five residues each side. The resulting samples had the uniform length of 22 amino acids, which partially covered two helices surrounding the central loop of the motif. Based on the spatial distance and the direct coupling analysis using Gremlin (*Ovchinnikov, Kamisetty & Baker, 2014*), only one pair of residues (between positions 8 and 17) was chosen for the training contact map. Redundancy reduction at level of sequence similarity of around 65% (using cd-hit) and pruning from corrupted sequences (due to

artifacts in pdb files) resulted in the sample of 37 sequences. (Later, it was discovered that a single false positive sequence was mistakenly included in the EF hand training set.)

*Grammatical descriptors.* Due to presumed higher complexity of the model, several variants of grammar rules were again used for training. The best fitting to the training sample was achieved with the same variant as in the previous example. Also in this case, learning with the contact constraints significantly improved fitness to the training data (probability mass distributed over the training set increased roughly 20 times on average over six runs).

The diagram showing the 36 most significant rules (all with probability of at least 0.05) and dependencies between structural non-terminals (possible derivations) of the single best grammar are shown in Fig. 5A. Of note is a pair of structural non-terminal symbols $u$ and $v$ (orange), which can be used to generate paired stretches of hydrophobic ($u \rightarrow ava$) and other residues ($v \rightarrow buc$). The feature was used to model the pair of beta-strands in the stem part of CaMn (Figs. 5B, 5C). By extending the cooperation between $u$ and $v$ with the derivation path through the structural non-terminal $t$ (pink, $v \rightarrow atb$, $t \rightarrow \bullet u \bullet$), the grammar generates hydrophobic residues with periodicity of 3, typical to helices, as used in modeling the pair of alpha-helices of the EF hand (Figs. 5D, 5E). To finish a derivation, it is typically necessary to use the structural non-terminal $w$ (green), which is likely to generate lexical non-terminals $b$ and $c$ which emit amino acids with high propensity to binding Calcium (aspartic and glutamic acids, aspargine, serine, and threonine (*Bindreither & Lackner, 2009*).

Clearly, the grammar has its limitations. The number of only three lexical non-terminals is likely insufficient, as suggested by the unusual merging of hydrophobic alanine with the charged amino acids in one group emitted through symbol $b$. Also detailed analysis of parse trees reveal inaccuracies possibly resulting from over-generalization. Most notably, the beta-hairpin generating rules (orange) were used to model a part of the binding loop of CaMn (Fig. 5B). Moreover, the residues directly involved in the Calcium binding in 1gsl, according to Ligplot (D130, W133, N135 and D140), were not generated with the non-terminal $w$. Finally, contact rules used to model the loop of the EF hand did not generate pairs of residues which are actually in contact. Yet, the overall topologies of the trees were rather consistent with the structures.

*Quantitative evaluation.* The grammar was used for scanning full sequences matching the EF hand and legume lectin Prosite patterns and profiles (PS00018, PS50222; PS00307) from the aforementioned set of the Calcium binding proteins (*Zhou, Tang & Altman, 2015*). Sequences with missing residues, non-canonical amino acid types and interfering ligands (except Manganese in the legume lectin set) were excluded. In 38 out of 40 sequences with the EF hands, and in all six sequences with the CaMn motif, the threshold of the log probability ratio of 3 between the grammar and the null model (meaning that a given sequence is 1,000 times more probable with the grammar than with the null) was exceeded in at least one position when scanned with the window ranging from 20 to 30 amino acids. In all EF hand and 5 CaMn hits, the highest score matched the position of the corresponding Calcium-binding Prosite motif (in one CaMn and one EF hand case

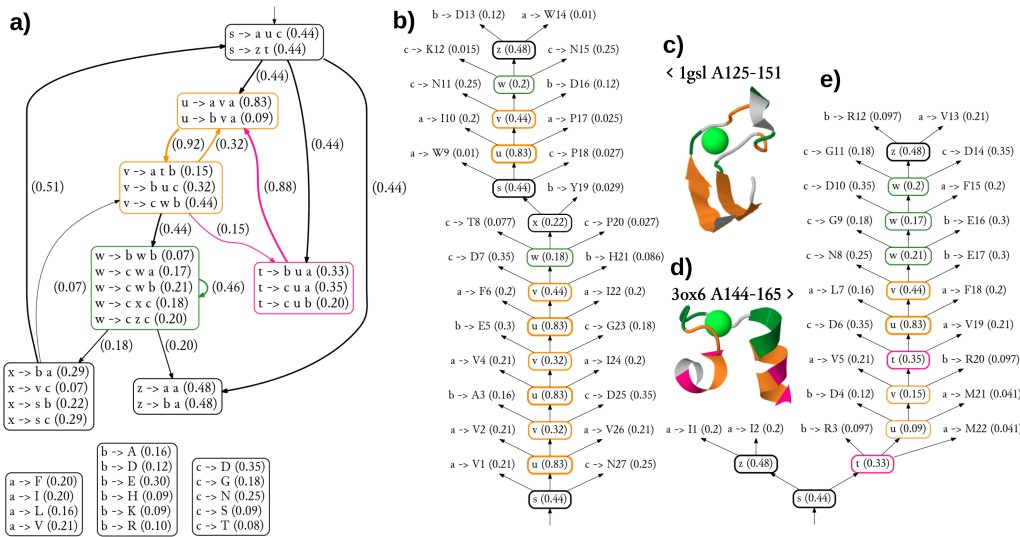

**Figure 5 Generalized grammar and parse trees for two calcium-binding motifs, the legume lectin CaMn motif and the EF hand.** (A) The diagram showing the 36 most significant rules (all with probability at least 0.05) and dependencies between structural non-terminals (possible derivations) of the single best grammar. Boxes with lexical rules are not connected for the sake of clarity. Colors indicate structural non-terminal symbols apparently used to model a pair of beta-strands (orange), a pair of helices (orange/pink), and the Calcium-binding loop (green). The graphical representation of the grammar has been partially inspired by *Unold, Kaczmarek & Culer (2017)*. (B) The most likely parse tree and (C) the cartoon structure of a highly scored training sequence from the CaMn family. (D) The cartoon structure and (E) the most likely parse tree of a highly scored training sequence from the EF hand family. Residue numbering is relative. Derivations of lexical symbols are represented using rules for the sake of brevity. Rule probabilities are shown in parentheses. Note that occasionally less probably rules, not shown in (A) are used. Colors correspond to structural non-terminals used to generate the residue according to the grammar. Structures were plotted using JSmol.

it was off center). In the remaining CaMn case, the highest score was at the position of another beta-loop-beta pair containing the characteristic alpha-chain signature PS00308. In terms of descriptive performance, the median average precision with regard to the full contact map was 0.23 for the EF hand and 0.65 for the legume lectin binding site using the sequence separation 3+ and the spatial distance cutoff of 8 Å. (The median AP increased to 0.43 and 0.72, respectively, for the distance cutoff of 10 Å.)

Eventually, the grammar was used to scan the representative set of all sequences in the PDB database at identity level of around 40% made with cd-hit (*Fu et al., 2012*) (25,145 sequences in total). Out of 48 hits which exceeded the log ratio of probability of six, the best matches in 15 sequences contained the low complexity regions made of stretches of amino acids with high affinity to binding Calcium (aspartic and glutamic acids, and asparagine). In the remaining part, 13 matches contained the PS00018 motif (out of 116 sequences with the motif in the set) and two matches contained the PS00307 motif (out of 18 in the set). In addition, experimental structures of four more sequences included the Calcium ion (out of 1,081 in the set), in three cases close to the grammar-defined match. To summarize, excluding matches to the low complexity fragments, there was an external

support for 18 out of 33 best hits in the scan with the grammar. Furthermore, assuming the log ratio of probability of three, candidate motifs were found in 4,419 sequences, including 114 matches to the low complexity regions, 72 matches to the PS00018 motif, five matches to the PS00307 motif and 340 matches to other Calcium-binding chains.

## DISCUSSION

### Added value of contact constraints

The primary evaluation of the PCFG-CM framework was conducted using samples of gapless alignments, which were based on datasets studied in our previous research (*Dyrka & Nebel, 2009*; *Daskalov, Dyrka & Saupe, 2015*) to limit potential confounding factors. (However, it has to be emphasized that, in general, training PCFG in our framework does not require alignment of sequences, as demonstrated in 'Making generalized descriptors). These initial tests focused on validating the proposed method for accommodating contact constraints in the training scheme for probabilistic context-free grammars.

The evaluation showed that the most effective way of training descriptors for a given sample was the contrastive estimation with reference to the contact map. This approach is only possible when a single contact map that fits all sequences in the target population can be used with the trained grammar. The maximum-likelihood estimators were effective when contacts were relevant to structure of the sequence (HET-s, CaMn). This is expected, as use of the contact rules is likely to be optimal for deriving a pair of amino acids in contact if they are actually correlated. Interestingly, in the case of HET-s, the maximum-likelihood grammar trained with the contact constraints compared favorably with the maximum-likelihood grammar trained without the constraints even when tested on sequences alone (AP 0.60 versus 0.43). This indicates that if contacts are relevant for the structure of sequence, the PCFG-CM approach can improve robustness of learning to local optima (similar effect was observed in both examples in 'Sample applications'). Of note is very good performance of grammars achieved for CaMn despite a tiny size of the positive set (18 training sequences in each fold), which can be attributed to high homogeneity of the sample (50% identity on average).

The most likely parse trees, derived for inputs defined only by sequences, reproduced a vast majority of contacts (recall of at least 0.79 at $\delta = 4$) enforced by the contact-constrained training input. Moreover, precision of contact prediction at $\delta = 4$ and sequence separation 3+ was above 0.50, up to 0.87. This translated to the overall overlap with the full contact maps in the range of 0.27–0.39. Note that only a fraction of contacts can be represented in the parse tree of context-free grammar, and not even all of them were enforced in training. The benefit of the contrastive estimation with reference to the sequence set was limited in comparison to the maximum-likelihood grammars. However, it should be noted that the shape of the most likely parse tree, which was used in the evaluation, does not necessarily reflect the most likely shape of parse tree. Unfortunately, the latter cannot be efficiently computed (*Dowell & Eddy, 2004*).

## Towards practical applications

The first experiments mainly served assessing intuitions which led to development of the PCFG-CM approach. The next task of searching the HET-s/r0 motifs showed good precision and recall, which indicated that in the current form our tool can be potentially useful for finding candidate sequences for further analysis in datasets of moderate sizes ('Searching for related motifs'). However, the average precision of evolved PCFGs was lower in comparison to profile HMMs. Therefore, improving specificity of the method is necessarily a premier goal for further research. The full-scale practical application to bioinformatic problems, such as sequence search, would certainly require several enhancements. This may include scoring inputs with the product of probabilities obtained using grammars with the lexical rule probabilities fixed according to representative physicochemical properties of amino acids (*Dyrka & Nebel, 2009*), and the appropriately adjusted null model to accurately account for various sequence lengths and amino acid compositions. In addition an extension of the PCFG-CM framework to account for uncertain contact information (*Knudsen, 2005*) can be obtained through introducing the concept of the fuzzy sets of syntactic trees.

The key challenge is, however, to enable learning grammars with increased number of non-terminal symbols. Currently implemented inference of rule probabilities using genetic algorithm worked well up to roughly half thousand rules, which translated to just a couple of non-terminal symbols for generic covering sets of rules. This necessarily imposed substantial level of generalization, which has advantages (simplicity of model and lower risk of over-fitting), but also drawbacks when the resulting grammar is too simple to capture complexity of the data. The low number of non-terminal symbols also effectively limits the length of modeled sequences, since longer fragments typically have more complex structures, which require more non-terminals to obtain a reasonable grammatical description. As the size of covering set of grammar rules is determined by the number of non-terminal symbols, therefore, the longer the sequence, the larger is the number of probabilities to be assigned. Sometimes, the problem can be partially overcome with generic constraints on the covering set of rules, as shown in sample applications ('Making generalized descriptors'). In this case, a meta-family of motifs was modeled using a grammar with 10 non-terminal symbols, which was trained starting from the constrained covering set of 675 rules. Yet, in general, more efficient estimation of probabilities of numerous rules and/or added capability of inferring rules during learning is required (*Unold, 2005*; *Unold, 2012*; *Coste, Garet & Nicolas, 2012*; *Coste, Garet & Nicolas, 2014*).

The potential of our approach beyond current state of the art was highlighted with the example of grammatical descriptor of a meta-family of Calcium binding sites. The PCFG evolved by our tool correctly generalized some common features of two distinctive folds and exhibited reasonable discriminative power. Both of the folds represented the loop-like structure, which can be modeled with the context-free grammar rules. As a result, parse trees generated by the grammar could directly correspond to the spatial structure of protein. However, it can be noted that every full graph of interactions can be decomposed to a set of trees consisting of the branching and nesting interactions. Thus, contact maps based on such trees can be used to train a set of context-free grammars, together covering

a large fraction of contacts. Another appealing solution is to modify the definition of consistency of the parse tree with the contact map, so that it requires that *only* residues in contact can be generated with the contact rules (instead of the definition used in this work that all residues in contact must be generated with the contact rules). The modified definition would allow using contact maps including crossing and overlapping contacts in the grammar learning. Indeed, multiple valid parse trees generated with the grammar for a sequence can potentially represent various branching and nesting subsets of dependencies. Nevertheless, the capability of capturing even only a fraction of non-local contacts, as in the current version of the framework, is already a step forward from the profile HMM, or probabilistic regular grammars.

## CONCLUSIONS

The complex character of non-local interactions between amino acids makes learning the languages of protein sequences challenging. In this work we proposed a solution consisting of using structural information to constrain syntactic trees, a technique which proved effective in learning probabilistic natural and RNA languages. We established a framework for learning probabilistic context-free grammars for protein sequences from syntactic trees partially constrained using contacts between amino acids. Within the framework, we implemented the maximum-likelihood and contrastive estimators for the rule probabilities of relatively simple yet practical covering grammars. Computational validation showed that additional knowledge present in the partial contact maps can be effectively incorporated into the probabilistic grammatical framework through the concept of a syntactic tree consistent with the contact map. Grammars estimated with the contact constraints maintained good precision when used as classifiers, and derived the most likely parse trees, displaying improved fidelity to protein structures compared to the baseline grammars estimated without the constraints.

Though tested in the learning setting consisting of optimizing only rule probabilities, the estimators defined in the present PCFG-CM framework can be used in more general learning schemes also inferring grammar structure. Indeed, such schemes may benefit even more from constraining their larger search space. It is also interesting to consider extending the framework beyond context-free grammars, as contacts in proteins are often overlapping and thus context-sensitive. In this case however, the one-to-one correspondence between the parse tree and the derivation breaks, therefore it may be advisable to redefine the grammatical counterpart of the spatial distance in terms of derivation steps in order to take advantage of higher levels of expressiveness.

## ACKNOWLEDGEMENTS

WD acknowledges Olgierd Unold for interesting discussions in the course of the project.

### Funding

This research has been funded by the National Science Centre, Poland (grant no 2015/17/D/ST6/04054) and was supported by the E-SCIENCE.PL Infrastructure. Hugo Talibart is funded by a PhD grant from the University of Rennes. Computational experiments have been partially carried out using resources provided by Wroclaw Centre for Networking and Supercomputing (http://wcss.pl) (grant no 98). There was no additional external funding received for this study. The funders had no role in study design, data collection and analysis, decision to publish, or preparation of the manuscript.

### Grant Disclosures

The following grant information was disclosed by the authors:
National Science Centre, Poland: 2015/17/D/ST6/04054.
E-SCIENCE.PL Infrastructure.
University of Rennes.
Wroclaw Center for Networking and Supercomputing: 98.

### Competing Interests

The authors declare there are no competing interests.

### Author Contributions

- Witold Dyrka conceived and designed the experiments, performed the experiments, analyzed the data, contributed reagents/materials/analysis tools, prepared figures and/or tables, authored or reviewed drafts of the paper, approved the final draft, elaborated the theoretical framework, wrote the software.
- Mateusz Pyzik contributed reagents/materials/analysis tools, wrote the software.
- François Coste and Hugo Talibart authored or reviewed drafts of the paper, elaborated the theoretical framework.

### Data Availability

   Source code of PCFG-CM is available at https://git.e-science.pl/wdyrka/pcfg-cm.

### Supplemental Information

Supplemental information for this article can be found online at http://dx.doi.org/10.7717/peerj.6559#supplemental-information.

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
