# Peer review of "Estimating probabilistic context-free grammars for proteins using contact map constraints"

_PeerJ, doi:10.7717/peerj.6559_

## Round 0.1 · original submission · Major Revisions

Both reviewers have concerns about the accessibility of your manuscript to a biological audience. Reviewer 2 also raises important concerns regarding the applicability of the methods to realistic protein sequences. Finally, I agree with Reviewer 1 that the code should be made publicly available.

·

Basic reporting

Major points:
1. It’s not immediately clear to me who the audience of this paper is. Though I’m interested and fairly well read in protein evolution, sequence-structure constraints, etc. there are a number of terms and concepts presented in this manuscript that are new to me and writing assumes that readers have a pretty far ranging prior background. While this of course may (and surely does to some extent) reflect limitations of this humble reviewer’s knowledge, the authors are nevertheless missing an opportunity to more clearly explain the value of this research more broadly to people who may be interested in these concepts (i.e. me). “Grammar” in general will of course be familiar to most but care should be taken to explain the usefulness of this concept to protein sequences/evolution. “Context-free grammar” on top of that is a term that many/most will likely have not heard of or will be only vaguely aware of. Other concepts such as “syntactic trees” and “parse trees” are just kind of thrown out there and I’m doubtful that most readers will immediately know what the authors are talking about. A toy example presented in a figure or two would really work wonders to teach the importance of these concepts and make the authors advances more clear and more widely applicable.

2. Of all the writing, the abstract in particular is dense and difficult to parse. This is going to be the “sales pitch” for most potential readers and I strongly encourage the authors to clean this up and simplify it with clear problem statements and proposed solutions.

3. There were a few typos/grammar issues throughout but it wasn’t anything too bad. That said, the abstract (in addition to being dense) had much worse English than the remainder of the manuscript and really starts out on the wrong note and should be thoroughly edited in addition to being simplified. A few examples throughout that I came across:

a. An overall poor start in terms of English/grammar is that the first sentence of the abstract is a bit unwieldy and I believe grammatically incorrect. “Learning language of protein…” == “Learning the language of protein…”.
b. Still in that first sentence of the abstract… having 2 “which” clauses in a sentence is clunky. This could be changed by giving a full sentence to defining context free grammars as I believe few readers will know/understand this term. It’s all to say that a sentence like this could slide by more freely in the middle of a manuscript but this reviewer is of the opinion that the abstract is the portion of the manuscript that should have the most careful editing / precision of language to hook potential readers
c. Also in the abstract “Within the framework…” == “Within this framework…”. “The” is a bit ambiguous as if there were one and only one whereas “this” specifies that we are discussing the authors contribution.
d. I’d avoid any mention of heavy concepts like parse trees in the abstract entirely if possible because these will need to be defined.
e. Page 2 lines 57/58 “which do not outperform significantly HMMs” == “which do not significantly outperform HMMs”
f. Page 12 line 346: “amino acid sequence of” == “amino acid sequences of”
g. Page 17 line 467: “Complex character…” == “The complex character…”
h. Page 17 line 497: “…may even more benefit…” == “…may benefit even more…”

4. Code should be made publicly available


Minor comments:
1. Figure 2: actual structures here would be nice in addition to the simplified diagrams

2. The mathematical abbreviations in Tables 2 and 3 headings could be simplified to increase readability. I had to keep flipping back to where these terms were defined to understand the differences in the columns. A more straightforward textual description might be easier to grasp for those who won’t be diving fully into the methods

3. Page 1 Line 28: “a near infinite number of sequences”. Perhaps it’s a mis-understanding on my behalf but I don’t see that an infinite number of sequences can exist, unless I suppose they are of infinite length but for a finite alphabet and bounded size it’s an of course astronomical and near infinite number but not infinite.

4. The introduction is very well written and seems to have been edited far more closely than the abstract. Though an orienting “summary” paragraph at the end of the introduction might be helpful to add to again focus the readers on the structure of the paper, the problem, the solution, etc.

5. The methods are extraordinarily dense (Pages 4-11). Which may suit some readers fine and I don’t think should necessarily be cut down. But I note it here only that this reviewer was unable to properly evaluate them and more generally would encourage the authors to make sure that their results are written predicated on the fact that few readers will have read / fully grasped their methods.

6. Conclusions/Discussion could probably be combined into a single section. Or more typically if both occur the conclusion is typically a shorter summary paragraph rather than the longer of the two. I thought that the conclusion section was well written, fairly addressed the limitations, and hypothesized paths forward for this research and this should probably be put in the discussion.

Experimental design

Major points:
1. I really struggled after two read-throughs to pin-point the problem that the authors were addressing. Learning the language of protein sequences seems to be the goal, but this is of course a very abstract goal and hard for the reader to grasp in practical terms. While working towards this goal, the authors do present results that are concrete and I was kind of surprised when they came given that this information is not clearly laid out in the abstract. A clear practical goal/application seems to be to use contact information to supplement sequence information in order to better discriminate protein family members but nowhere is this made entirely clear. Being able to better define protein families using these methods could for instance have a large practical benefit of increasing the accuracy of protein homology search (a very valuable application). Page 2 lines 37-53 are the clearest articulation of this vision in the context of broader research but this definition of the problem appears nowhere in the abstract and really seems limited in the overall arc of the paper. If I’m correct that this seems to be the main practical benefit/goal of such research, this really needs to be made more explicit and scattered throughout the abstract/intro/results/discussion. If this is one of the main practical goals, well then it would seem apt to compare the methods developed here to the more limited profile-HMMs in the results (or are profile-HMMs fully equivalent to the grammars estimated without contact constraints? Entirely possible that this is the case, but if so, I think that’s another point of confusing terminology as I believe the term profile-HMM will be far more widespread to the likely readers of this paper than context-free/-sensitive grammars. The solution would simply to be to reiterate the fact that those methods are equivalent, but I’m still not sure that they are).

2. I still don’t fully get the “Descriptive power” section. Which again just relates to some lack of clarity in the writing or assumption of prior knowledge that I simply do not have. What precisely is even being classified/described here with the precision scores? True positive contacts from the structure? It’s not clear how their method converts probabilities into binary predicted contacts. Which is to say, it might be described there but again in a very abstract way that I have difficulty extracting using their terminology. It seems that this information comes directly from parse trees but if the link is kind of clear and obvious well then perhaps a diagram or figure again could show this link to someone who is unfamiliar with some of these concepts (i.e. has never heard of a parse tree, as I suspect some/many readers may be in a similar situation to myself). Additionally the value of that “descriptive power” problem seems a bit weak. Which is to say training a model with contact information helps better predict contacts? I’m not sure if I’m getting all of that properly, but it seems a bit tautological and not entirely surprising. The authors should clarify/expand why I’m being unfair in that regard.


Minor comments:
1.Table 1: It’s unclear to me why the number of nneg varies for the different protein fragment targets. Is this a consequence of the different lengths? 829 negative sequences were used, and it is said that these are cut into matching lengths for the positive set (are these cuts overlapping? i.e. is a 100amino acid sequence with target length 20 cut into 5 sequences or 80?) so this would make sense but could be made a bit more methodologically explicit.

Validity of the findings

No comment.

Additional comments

Dyrka et al. approach an important and topical issue, the relationship between structural constraints and protein sequences, using the concepts of context-free grammars. Notably, they show how to incorporate protein contact constraints into context-free grammars to develop methods that can better discriminate protein family members and re-capitulate the structural properties of the protein families in question. Overall, I find that the research is solid and can think of few/no actual analytical objections to any of the research presented that would require any re-analysis or further experiments. However, the biggest qualms I have with the paper are that it presents a very high bar of assumed prior knowledge and a lack of clear goals that severely limits comprehension of their results and advances.

Reviewer 2 ·

Basic reporting

The authors present the description of sequences of protein fragments using probabilistic context-free grammars (PCFG) using contact map constraints. This approach is successful in statistical models of RNA since the contacts in RNA secondary structure naturally fit the framework of PCFG (nested but no crossing contacts). These methods have not been applied much to proteins, which are more frequently described via profile HMM. The latter cannot include non-local interactions, and thus structural constraints for contacts bringing together residues distant in the primary structure. The authors propose PCFG to overcome these limitations.

While the generalization beyond HMM is an important question, I have a number of basic remarks concerning the manuscript. In general, it is written quite clearly, but also in a very formal language accessible mostly to computer scientists. It seems more written for the proceedings of a computer science conference than for a life-science journal. An effort to give motivations behind formal constructions should be made.

Experimental design

se below under "Validity of findings"

Validity of the findings

1) The authors propose PCFG as structurally informed probabilistic models for protein sequences. I do not understand the motivations for this choice, since contact maps of proteins have many characteristics, which do not seem compatible with the construction rule of CFG: Crossing contacts like in alpha helices (e.g. (i,i+4) and (i+1,i+5)) or parallel beta sheets (e.g. (i,j), (i+1,j+1),…, (i+l),(j+l)), residues with multiple contacts two distinct regions of the protein instead of one-to-one contacts. So the target might be small fragments like the ones shown in Fig. 2 and analyzed in the paper (by the way, Fig. 2 is basically unreadable, a little graphical effort might be helpful there). But wouldn’t the effort of developing PCFG be large for fragments of very specific structure? I miss an overall motivation and justification of the model setup, and a serious discussion of the limitations of application. More general graphical models or Markov random fields (MRF) seem more adapted to the complex structure of protein contact maps.

2) In the definition of te PCFG used, it remains unclear why V_T is not directly identified with \Sigma? Probably the use of three symbols for V_T ans four for V_N makes the model more flexible (more parameters) than PCFG used for RNA, but why values 3 and 4 for these cardinalities? The justification seems to be the computational complexity of the problem, but is there any argument that less symbols would be worse in performance? Also profile HMM do not have this kind of multiplicity, since each match state has a single amino-acid emission matrix. The precise model settings remain thus somehow unmotivated.

3) Even if the number of elements of V_T, V_N remains restricted, the PCFG framework currently does not seem to be applicable to sequences longer than 20-30 amino acids, and alignments have to be gapless. This seems quite a restriction. MRF are currently used for hundreds of amino acids in multiple-sequence alignments with gaps.

4) In the tests on protein fragments, very few positive sequences (24-160) are used. Why so few? The fact that performance decreases with npos is counter-intuitive: One would expect that model parameters are more precisely estimated when samples are larger. Larger samples would also, in principle, make average-precision results less noisy.

5) The entire protein space is approximated by 829 single sequences of length 300-500, constructed from old data in 2006. Uniprot contains more than 100 million sequences, Pfam lists more than 16,000 protein families, the majority of them containing structurally resolved examples in the PDB. So I would guess that the negative set is an extremely rough approximation of protein sequence space, but the resulting sample is already three orders of magnitude larger than the corresponding npos values.

Additional comments

I am afraid that I miss some basic point, but even after reading the manuscript several times I continue to miss it. I would therefore suggest the authors to substantially revise their manuscript to make their work more accessible and the motivations and limitations – in particular for biological applications – more clear.

---

## Round 0.2 · Minor Revisions

Both reviewers appreciate the extensive revisions you have performed. However, one reviewer feels the manuscript remains fairly inaccessible to a broader biological audience. I agree with this assessment, and I would like to ask you to go over the manuscript one more time and see if there are ways to make the presentation more accessible.

I will leave the extent to which you want to carry out such revisions up to you. I agree with the 2nd reviewer that it is in your interest to write an article that is broadly accessible and has an impact on its field. I do not expect that a revision will have to be re-reviewed.

·

Basic reporting

In my initial review I raised a number of concerns about the writing style and presentation of the results, finding it to initially be a very dense paper for a biological audience. In this revision, the authors have completely overhauled the manuscript. In the current version, the writing is clear, the problem is introduced well, the conclusions are clear, software is made available (with a working link), and the figures (particularly the addition of a toy example) are much improved. I see no other major concerns to note at this point

Experimental design

No comment.

Validity of the findings

No comment.

Additional comments

Responses to my initial comments/concerns were both very thorough and courteous. I believe that the revision to be a much improved manuscript, and thank the authors for taking the time and care to address those initial concerns so clearly and comprehensively.

Reviewer 2 ·

Basic reporting

In their revised version, the authors have made a major effort to improve the presentation of their paper. However, as pointed already out by both reviewers in the first round, the paper remains quite hard to access:

- It contains valid scientific work, even if the application to small protein fragments appears limiting as compared to the formal development beyond the PCFG-CM approach.
- The presentation is very formal. Much of the notations is far beyond what is needed to follow the manuscript, but they make the article inaccessible to a large fraction of the potentially interested readership (e.g. researchers interested in protein sequence modeling and annotation).
- The discussion of the sample applications in terms of, e.g., discriminative power and average precision remains quite superficial, concentrating on the comparision of a few numbers characterizing global performance.
- Some of my questions have been answered in the rebuttal letter, but these answers have not necessarily found a clear way into the manuscript (e.g. the selection of data sets, scanning e.g. SwissProt or PDB with the Prosite motifs gives much more positive hits; the selection of some model details like the cardinalities of VN and VT etc). The motivation and generality behind these choices remains thus unclear in the paper.

So the impression is that the almost the entire interest of the authors went into the formal development of the PCFG-CM framework, and little into the biological problem. From my point of view (but I might be the wrong reviewer for this kind of manuscript, not having a background in computer science), this is a pitty. Early papers suggesting very similar mathematical structures - PCFG with structural constraints – written by authors like Eddy, Durbin, Haussler and others in the 1990s, are much more accessible to a broader audience. These papers have, without doubt, changed our way to bioinformatically look at RNA.

Experimental design

-

Validity of the findings

-

Additional comments

I sincerely think that a less formal presentation of the material would facilitate the access to this work. However, the style of presentation should be a choice of authors.

---

## Author Rebuttal · Round 0.2

Dear Editor,

we would like to thank you for considering our manuscript, and thank the reviewers for their fair and insightful comments. The manuscript has been substantially updated to address the suggestions. We am convinced that it is now far more appealing to a biological audience.

Following your suggestion we make the source code of the project public. To give access to a clean code, the source code underwent almost complete and non-trivial refactoring, which was mostly performed by the member of my team Mateusz Pyzik, who therefore deserves authorship of the manuscript.

We believe that the manuscript is now suitable for publication in PeerJ.

Yours sincerely,
Witold Dyrka

*Reviewer 1:*

*Basic reporting*

*Major comments:*

*1. It's not immediately clear to me who the audience of this paper is. Though I'm interested and fairly well read in protein evolution, sequence-structure constraints, etc. there are a number of terms and concepts presented in this manuscript that are new to me and writing assumes that readers have a pretty far ranging prior background. While this of course may (and surely does to some extent) reflect limitations of this humble reviewer's knowledge, the authors are nevertheless missing an opportunity to more clearly explain the value of this research more broadly to people who may be interested in these concepts (i.e. me). "Grammar" in general will of course be familiar to most but care should be taken to explain the usefulness of this concept to protein sequences/evolution. "Context-free grammar" on top of that is a term that many/most will likely have not heard of or will be only vaguely aware of. Other concepts such as "syntactic trees" and "parse trees" are just kind of thrown out there and I'm doubtful that most readers will immediately know what the authors are talking about. A toy example presented in a figure or two would really work wonders to teach the importance of these concepts and make the authors advances more clear and more widely applicable.*

We followed the suggestion of the Reviewer and added a figure presenting a toy example aimed at illustrating usefulness of grammatical descriptors to protein sequences, and the concepts of probabilistic context-free grammar, derivation, parse tree and contact map.

*2. Of all the writing, the abstract in particular is dense and difficult to parse. This is going to be the "sales pitch" for most potential readers and I strongly encourage the authors to clean this up and simplify it with clear problem statements and proposed solutions.*

The abstract was completely rewritten in line with suggestions of the Reviewer.

*3. There were a few typos/grammar issues throughout but it wasn't anything too bad. That said, the abstract (in addition to being dense) had much worse English than the remainder of the manuscript and really starts out on the wrong note and should be thoroughly edited in addition to being simplified. A few examples throughout that I came across:*

*a.      An overall poor start in terms of English/grammar is that the first sentence of the abstract is a bit unwieldy and I believe grammatically incorrect. "Learning language of protein…" == "Learning the language of protein…".*

*b.      Still in that first sentence of the abstract… having 2 "which" clauses in a sentence is clunky. This could be changed by giving a full sentence to defining context free grammars as I believe few readers will know/understand this term. It's all to say that a sentence like this could slide by more freely in the middle of a manuscript but this reviewer is of the opinion that the abstract is the portion of the manuscript that should have the most careful editing / precision of language to hook potential readers*

*c.      Also in the abstract "Within the framework…" == "Within this framework…". "The" is a bit ambiguous as if there were one and only one whereas "this" specifies that we are discussing the authors contribution.*

None of these incorrect phrases is present in the rewritten abstract.

*d.       I'd avoid any mention of heavy concepts like parse trees in the abstract entirely if possible because these will need to be defined.*

We rewritten the abstract in accordance to the suggestion.

*e.      Page 2 lines 57/58 "which do not outperform significantly HMMs" == "which do not significantly outperform HMMs"*
*f.       Page 12 line 346: "amino acid sequence of" == "amino acid sequences of"*
*g.      Page 17 line 467: "Complex character…" == "The complex character…"*
*h.      Page 17 line 497: "…may even more benefit…" == "…may benefit even more…"*

We corrected the manuscript according to the suggestions.

*4. Code should be made publicly available*

The source code has been made available in the git repository at https://git.e-science.pl/wdyrka/pcfg-cm under the GPL 3 license. The source code underwent almost complete and non-trivial refactoring. The functional compatibility with the previous version was controlled through acceptance tests.

*Minor comments:*

*1. Figure 2:  Actual structures here would be nice in addition to the simplified diagrams.*

We agree with the Reviewer. In the revised manuscript, the figure presents backbones of the actual structures (with some annotations) instead of the simplified diagrams.

*2. The mathematical abbreviations in Tables 2 and 3 headings could be simplified to increase readability. I had to keep flipping back to where these terms were defined to understand the differences in the columns. A more straightforward textual description might be easier to grasp for those who won't be diving fully into the methods*

We dropped the mathematical abbreviations from the table headers in favor of the textual descriptions and abbreviations.

*3. Page 1 Line 28: "a near infinite number of sequences". Perhaps it's a mis-understanding on my behalf but I don't see that an infinite number of sequences can exist, unless I suppose they are of infinite length but for a finite alphabet and bounded size it's an of course astronomical and near infinite number but not infinite.*

We thank the Reviewer for the remark. We agree that number of protein/nucleic acid sequences is not infinite since they are limited in length and alphabet. On the other hand, formal languages with an infinite number of sentences exist. Therefore, in the updated manuscript we opted for the word "enormous" instead of "infinite" to account for this difference.

*4. The introduction is very well written and seems to have been edited far more closely than the abstract. Though an orienting "summary" paragraph at the end of the introduction might be helpful to add to again focus the readers on the structure of the paper, the problem, the solution, etc.*

We thank the Reviewer for the comment and the helpful suggestion. The section entitled "Contributions of this research" was added at the end of the introduction.

*5. The methods are extraordinarily dense (Pages 4-11). Which may suit some readers fine and I don't think should necessarily be cut down. But I note it here only that this reviewer was unable to properly evaluate them and more generally would encourage the authors to make sure that their results are written predicated on the fact that few readers will have read / fully grasped their methods.*

We thank Reviewer for the comment. We believe that with all extensions made to the manuscript, it is far more accessible to a broader community.

*6. Conclusions/Discussion could probably be combined into a single section. Or more typically if both occur the conclusion is typically a shorter summary paragraph rather than the longer of the two. I thought that the conclusion section was well written, fairly addressed the limitations, and hypothesized paths forward for this research and this should probably be put in the discussion.*

In fact, it was the editorial team of the journal who required us to split "Discussion" and "Conclusions". In the current version, we extended the "Discussion" section and improved arrangement of the content between the both.

*Experimental design*

*Major points:*

*1. I really struggled after two read-throughs to pin-point the problem that the authors were addressing. Learning the language of protein sequences seems to be the goal, but this is of course a very abstract goal and hard for the reader to grasp in practical terms. While working towards this goal, the authors do present results that are concrete and I was kind of surprised when they came given that this information is not clearly laid out in the abstract. A clear practical goal/application seems to be to use contact information to supplement sequence information in order to better discriminate protein family members but nowhere is this made entirely clear. Being able to better define protein families using these methods could for instance have a large practical benefit of increasing the accuracy of protein homology search (a very valuable application). Page 2 lines 37-53 are the clearest articulation of this vision in the context of broader research but this definition of the problem appears nowhere in the abstract and really seems limited in the overall arc of the*

*paper. If I'm correct that this seems to be the main practical benefit/goal of such research, this really needs to be made more explicit and scattered throughout the abstract/intro/results/discussion. If this is one of the main practical goals, well then it would seem apt to compare the methods developed here to the more limited profile-HMMs in the results (or are profile-HMMs fully equivalent to the grammars estimated without contact constraints? Entirely possible that this is the case, but if so, I think that's another point of confusing terminology as I believe the term profile-HMM will be far more widespread to the likely readers of this paper than context-free/-sensitive grammars. The solution would simply to be to reiterate the fact that those methods are equivalent, but I'm still not sure that they are).*

We modified and extended several parts of the manuscript in order to improve presentation of the motivation and the goal of this piece of research. First, we put them forward briefly in the abstract: "Interactions between amino acids that are close in the spatial structure, but not necessarily in the sequence, play important structural and functional roles in proteins. These non-local interactions ought to be taken into account when modeling collections of proteins. Yet, the most popular representations of sets of related protein sequences remain the profile Hidden Markov Models. Modeling independently the distributions of the conserved columns from an underlying multiple sequence alignment of the proteins, they are are unable to capture dependencies between the protein residues. Non-local interactions can be represented using more expressive grammatical models. However, learning such grammars is difficult. In this work, we propose to use information of protein contacts to facilitate training probabilistic context-free grammars for protein sequences. We develop the theory behind this new estimation scheme and implemented it in a machine learning framework for protein grammars."

Second, we updated the introduction to concisely present limitations of profile HMMs, probabilistic regular grammars and the Potts model in modeling protein sequences. This motivates our research towards application of the probabilistic context-free grammar:
"The architecture of a profile HMM corresponds to the underlying multiple sequence alignment (MSA). Thus, the model perfectly suits modeling single-point mutations and supports insertions and deletions, but cannot account for interdependence between positions in the MSA. Pairwise correlations in a MSA can be statistically modeled by extended Potts model (a type of Markov Random Field or undirected graphical model). This has been highly successful to predict 3D contact between residues of a protein, but computing the probability of new (unaligned) sequences with such model is untractable. An alternative to MSA-based modeling, is to use formal grammars. Protomata are probabilistic regular models that can capture local dependencies. Yet, as regular level models, they are not well suited to capture the interactions occurring between amino acids which are distant in sequence but close in the spatial structure of the protein. In that case, formal grammars beyond regular are needed. Specifically, the context-free (CF) grammars are able to represent interactions producing nested and branched dependencies (an example is given in Fig. 1), while the context-sensitive (CS) grammars can also represent overlapping and crossing dependencies. The sequence recognition problem is untractable for CS grammars, but it is polynomial for CF and mildly context sensitive grammars."

Third, we explicitly state the goal and define the contributions of this research in the newly added "Contributions of this research" section:
"In the broader plan, this research aims at developing a protein sequence analysis method advancing the current state of the art represented by the profile HMMs in being not limited to alignment-defined protein sequence families, and capable of capturing of interactions between amino acids. The ideal approach would be based on the probabilistic (mildly) context-sensitive grammars, however their computational complexity significantly hampers practical solutions. Therefore, an intermediate approach based on the probabilistic context-free grammars is considered here, which is computationally cheaper and can represent the non-crossing (and non-overlapping) interactions

between amino acids. Still, the main difficulty is efficient estimation of the grammars. Our solution is to accommodate information of protein contacts as structural constraints for the model estimation and, if possible, for the sequence analysis. The first contribution of this work consists on developing a theoretical framework for defining the maximum-likelihood and contrastive estimators of PCFG using contact constraints (section 2.1). Building on this general framework, the second contribution of this work is extension of our previous probabilistic context-free grammatical model for protein sequences (Dyrka 2007, Dyrka and Nebel 2009, Dyrka et al. 2013), proposed in section 2.2."

Finally, in the added "Sample applications" subsection of "Results", we present an example of using the method in a practical setting (where performance of the PCFG is also compared with the profile HMMs) and demonstrate the potential of our approach beyond the current state of the art by creating a grammatical model of a meta-family of protein motifs. These results are then commented in the "Towards practical applications" section of "Discussion".

*2. I still don't fully get the "Descriptive power" section. Which again just relates to some lack of clarity in the writing or assumption of prior knowledge that I simply do not have. What precisely is even being classified/described here with the precision scores? True positive contacts from the structure? It's not clear how their method converts probabilities into binary predicted contacts. Which is to say, it might be described there but again in a very abstract way that I have difficulty extracting using their terminology. It seems that this information comes directly from parse trees but if the link is kind of clear and obvious well then perhaps a diagram or figure again could show this link to someone who is unfamiliar with some of these concepts (i.e. has never heard of a parse tree, as I suspect some/many readers may be in a similar situation to myself). Additionally the value of that "descriptive power" problem seems a bit weak. Which is to say training a model with contact information helps better predict contacts? I'm not sure if I'm getting all of that properly, but it seems a bit tautological and not entirely surprising. The authors should clarify/expand why I'm being unfair in that regard.*

In this piece of work, we define the "descriptive performance" as "the capability of acquiring contact constraints by the grammar" (in the "Contributions of this research" section) and explain in detail in the substantially enhanced relevant paragraph of the "Performance measures" section: "Intuitively, a decent explanatory grammar generates parse trees consistent with the spatial structure of the analyzed protein. Therefore, the descriptive performance of grammar can be quantified as the amount of contact information encoded in the grammar and imposed on its derivations. In other words, it is expected that the grammar ensures that residues in contact are close in the parse tree. The most straightforward approach to measure the descriptive performance is to use the skeleton of the most likely parse tree as a predictor of spatial contacts between positions in a given protein sequence, parameterized by the cutoff delta on path length between the leaves. The natural threshold for grammar in the CFC form is delta=4 meaning that the pair of residues is predicted to be in contact if they are parsed with a contact rule. The precision at this threshold was reported for CFC grammars since the precision is the usual measure of contact prediction performance. In addition, AP of the RPC, which sums up over all possible cutoffs, was computed to allow comparison with grammars without pairing rules. Our recent research suggests that the measure is suitable for the contact-map-based comparison of the overall topology of parse trees generated with various grammars (Pyzik et al. 2018, in press). Since our definition of consistency between the parse tree and the contact map imposes that inferred grammars maximize the recall rather than the precision of contact prediction, the learning process was assessed using the recall measured with regard to the partial contact map used in the training for delta=4. (...)"

Our recent manuscript on the topic will appear in the proceedings of the 14th International Conference on Grammar Inference ICGI2018 (Proceedings of Machine Learning Research, vol 93, in press).

We also added Fig. 4 and 5 presenting sample parse trees, in addition to Fig. 1 presenting parse trees for a toy example.

*Minor comments:*

*1. Table 1: It's unclear to me why the number of nneg varies for the different protein fragment targets. Is this a consequence of the different lengths? 829 negative sequences were used, and it is said that these are cut into matching lengths for the positive set (are these cuts overlapping? i.e. is a 100amino acid sequence with target length 20 cut into 5 sequences or 80?) so this would make sense but could be made a bit more methodologically explicit.*

Indeed, this is due to the different lengths of positive sequences. The negative sequences were cut into overlapping fragments of the length of positive sequences, which is now explicit in the text.

*Validity of the findings*

*No comment.*

*Comments for the Author*

*Dyrka et al. approach an important and topical issue, the relationship between structural constraints and protein sequences, using the concepts of context-free grammars. Notably, they show how to incorporate protein contact constraints into context-free grammars to develop methods that can better discriminate protein family members and re-capitulate the structural properties of the protein families in question. Overall, I find that the research is solid and can think of few/no actual analytical objections to any of the research presented that would require any re-analysis or further experiments. However, the biggest qualms I have with the paper are that it presents a very high bar of assumed prior knowledge and a lack of clear goals that severely limits comprehension of their results and advances.*

We would like to thank the Reviewer for assessing our manuscript. We are convinced that with the help of the comments, we significantly improved presentation of our work.

*Reviewer 2*

*Basic reporting*

*The authors present the description of sequences of protein fragments using probabilistic context-free grammars (PCFG) using contact map constraints. This approach is successful in statistical models of RNA since the contacts in RNA secondary structure naturally fit the framework of PCFG (nested but no crossing contacts). These methods have not been applied much to proteins, which are more frequently described via profile HMM. The latter cannot include non-local interactions, and thus structural constraints for contacts bringing together residues distant in the primary structure. The authors propose PCFG to overcome these limitations.*

*While the generalization beyond HMM is an important question, I have a number of basic remarks concerning the manuscript. In general, it is written quite clearly, but also in a very formal language accessible mostly to computer scientists. It seems more written for the proceedings of a computer science conference than for a life-science journal. An effort to give motivations behind formal constructions should be made.*

We would like to thank the Reviewer for assessing our manuscript. We followed the suggestions of the reviewer and substantially revised the manuscript to make it accessible to a life-science audience. This includes the completely rewritten abstract which presents motivation and contributions as follows:

"Interactions between amino acids that are close in the spatial structure, but not necessarily in the sequence, play important structural and functional roles in proteins. These non-local interactions ought to be taken into account when modeling collections of proteins. Yet, the most popular representations of sets of related protein sequences remain the profile Hidden Markov Models. Modeling independently the distributions of the conserved columns from an underlying multiple sequence alignment of the proteins, they are are unable to capture dependencies between the protein residues. Non-local interactions can be represented using more expressive grammatical models. However, learning such grammars is difficult. In this work, we propose to use information of protein contacts to facilitate training probabilistic context-free grammars for protein sequences. We develop the theory behind this new estimation scheme and implemented it in a machine learning framework for protein grammars."

*Experimental design*

*see below under "Validity of findings"*

*Validity of the findings*

*1) The authors propose PCFG as structurally informed probabilistic models for protein sequences. I do not understand the motivations for this choice, since contact maps of proteins have many characteristics, which do not seem compatible with the construction rule of CFG: Crossing contacts like in alpha helices (e.g. (i,i+4) and (i+1,i+5)) or parallel beta sheets (e.g. (i,j), (i+1,j+1),..., (i+l),(j+l)), residues with multiple contacts two distinct regions of the protein instead of one-to-one contacts. So the target might be small fragments like the ones shown in Fig. 2 and analyzed in the paper (by the way, Fig. 2 is basically unreadable, a little graphical effort might be helpful there). But wouldn't the effort of developing PCFG be large for fragments of very specific structure? I miss an overall motivation and justification of the model setup, and a serious discussion of the limitations of application. More general graphical models or Markov random fields (MRF) seem more adapted to the complex structure of protein contact maps.*

We thank the Reviewer for the comment.

In the updated manuscript, we write explicitly that "[i]n the broader plan, this research aims at developing a protein sequence analysis method advancing the current state of the art represented by the profile HMMs in being not limited to alignment-defined protein sequence families, and capable of capturing of interactions between amino acids." This is a very ambitious goal, and "[t]he ideal approach would be based on the probabilistic (mildly) context-sensitive grammars, however their computational complexity significantly hampers practical solutions". Thus, we chose to approach the big target in the step by step manner, and "[t]herefore, [we opted for] an intermediate approach based on the probabilistic context-free grammars (...), which is computationally cheaper and can represent the non-crossing (and non-overlapping) interactions between amino acids" (in the "Contributions of this research" section).

The first contribution of this piece of research, which consists on developing a theoretical framework for defining the maximum-likelihood and contrastive estimators of PCFG using contact constraints (section 2.1) may likely be extensible to the case of mildly context-sensitive grammars (as pointed out in the end of "Conclusions").

Nevertheless, the capability of capturing even only a fraction of non-local contacts is still a step forward from the profile HMM or probabilistic regular grammars. Moreover, "it can be noted that every full graph of interactions can be decomposed to a set of trees consisting of the branching and nesting interactions. Thus, contact maps based on such trees can be used to train a set of context-free grammars, together covering a large fraction of contacts. Another appealing solution is to modify the definition of consistency of the parse tree with the contact map, so that it requires that only residues in contact can be generated with the contact rules (instead of the definition used in this work that all residues in contact must be generated with the contact rules). The modified definition would allow using contact maps including crossing and overlapping contacts in the grammar learning. Indeed, multiple valid parse trees generated with the grammar for a sequence can potentially represent various branching and nesting subsets of dependencies." (as noted in the enhanced "Discussion" section).

There are alternatives to grammatical modeling, and the approach of choice may depend on particular applications. In the enhanced "Introduction" section, we briefly present limitations of the profile HMM, probabilistic regular grammars and the Potts model in modeling protein sequences: "The architecture of a profile HMM corresponds to the underlying multiple sequence alignment (MSA). Thus, the model perfectly suits modeling single-point mutations and supports insertions and deletions, but cannot account for interdependence between positions in the MSA. Pairwise correlations in a MSA can be statistically modeled by extended Potts model (a type of Markov Random Field or undirected graphical model). This has been highly successful to predict 3D contact between residues of a protein, but computing the probability of new (unaligned) sequences with such model is untractable. An alternative to MSA-based modeling, is to use formal grammars. Protomata are probabilistic regular models that can capture local dependencies. Yet, as regular level models, they are not well suited to capture the interactions occurring between amino acids which are distant in sequence but close in the spatial structure of the protein. In that case, formal grammars beyond regular are needed. Specifically, the context-free (CF) grammars are able to represent interactions producing nested and branched dependencies (an example is given in Fig. 1), while the context-sensitive (CS) grammars can also represent overlapping and crossing dependencies. The sequence recognition problem is untractable for CS grammars, but it is polynomial for CF and mildly context sensitive grammars."

The Fig. 2 was improved as requested by the Reviewer.

*2) In the definition of te PCFG used, it remains unclear why V_T is not directly identified with \Sigma? Probably the use of three symbols for V_T ans four for V_N makes the model more flexible (more parameters) than PCFG used for RNA, but why values 3 and 4 for these cardinalities? The justification seems to be the computational complexity of the problem, but is there any argument that less symbols would be worse in performance? Also profile HMM do not have this kind of multiplicity, since each match state has a single amino-acid emission matrix. The precise model settings remain thus somehow unmotivated.*

The number of three lexical non-terminal was assumed from our previous research (Dyrka and Nebel 2009), in which lexical rule probabilities were fixed according to representative physicochemical properties of amino acids. In (Dyrka and Nebel 2009) it seemed justified to have distinct symbols for the low, medium and high levels of the properties. The previous research used also four structural non-terminals as the baseline in order to limit the total number of possible rules in the Chomsky Normal Form. In the updated version of the manuscript the "Evaluation" subsection of the methodology was rewritten to explain the issue:
"The present approach for learning PCFGs with the contact constraints was evaluated using our evolutionary framework for learning the probabilities of rules. The underlying non-probabilistic CFGs were based on grammars used in our previous research (Dyrka and Nebel 2009), (...) For the

sake of transparent evaluation, combinations of symbols in the rules were not constrained beyond general definition of the CNF or CFC model, respectively, to avoid interference with the contact constraints. The number of non-terminal symbols was limited to a few in order to keep the number of parameters to be optimized by the genetic algorithm reasonably small. The small number of non-terminals implied relatively high generality of the resulting model, for example, only three distinct emission profiles of amino acids were defined by the lexical rules. Clearly, this has to be expected to confine specificity and limit attainable discriminatory power of the grammars. Although adjusting proportion of lexical and structural non-terminals could potentially improve performance of the grammatical model, it was not explored here, since the focus of evaluation was on the added value of the contact constraints for learning rule probabilities, rather than on the optimal set of rules."

Interestingly, when referring to the profile HMM, each match state can be identified with a lexical non-terminal and emissions in the state with, a set of lexical rules for this (left-hand-side) non-terminal.

*3) Even if the number of elements of V_T, V_N remains restricted, the PCFG framework currently does not seem to be applicable to sequences longer than 20-30 amino acids, and alignments have to be gapless. This seems quite a restriction. MRF are currently used for hundreds of amino acids in multiple-sequence alignments with gaps.*

It has to be emphasized that training PCFG in our framework does not require alignment of sequences. For evaluation of the estimation scheme with contact constraints, we chose to use datasets we knew well from our previous research to avoid confounding factors. We used the aligned and gapless sets because we wanted to avoid length effects on probabilities of sequences given a grammar, as described in the "Basic evaluation" section: "Within each sample, all sequences shared the same length, which avoided sequence length effects on grammar scores". In the revised manuscript, we show also an example of training PCFG for a combined set of two types of motifs, which are hardly alignable with each other.

Regarding practical constraints on the sequence length, there are two aspects. First is the cubic parsing complexity with regard to the sequence length. We checked that the training of sequence of the length close to 50 amino acids could last for about 100h on 12 modern cores. This is long but not prohibitive. Moreover, we checked that parsing an entire bacterial genome (about 5000 proteins) with a semi-automatically created grammar and using the scanning window of over 100 amino acids took about 3h on 4 cores working in the background. Again, this seems acceptable.

The second aspect is that longer fragments typically have more complex structures, which require more non-terminals to obtain a reasonable grammatical description. Since the size of covering set of grammar rules is determined by the number of non-terminal symbols, therefore, the longer the sequence, the larger is the number of probabilities to be assigned. In sample applications described in this updated manuscript, we were able to train appealing grammars with 10 non-terminal symbols, which translated to the total of 675 rules thanks to some generic constraints on the covering set of rules (as described in section "Sample applications"). Nevertheless, we believe that "the key challenge is to enable learning grammars with increased number of non-terminal symbols. This includes more efficient estimation of probabilities of numerous rules and/or added capability of inferring rules during learning", as stated in the "Towards practical applications" section of "Discussion".

*4) In the tests on protein fragments, very few positive sequences (24-160) are used. Why so few? The fact that performance decreases with npos is counter-intuitive: One would expect that model*

*parameters are more precisely estimated when samples are larger. Larger samples would also, in principle, make average-precision results less noisy.*

We thank the Reviewer for the comment. The numbers resulted from construction of the sets. Initially the sets included all sequences available at the time based on the Prosite website for CaMn and NAP, or on our previous publication for HET-s. Only small fractions were dropped because of atypical lengths. More reduction resulted from reducing redundancy to the similarity threshold of around 70%. In subsection "Materials" of "Basic evaluation", we explain that "[t]he threshold balanced the size of positive samples, distribution of their variability, and inter-fold diversity. Overall diversity of samples ranged from the most homogeneous CaMn (average identity of 49%) to the most diverse HET-s, which consisted of 5 subfamilies (average identity of 21%)." In the updated discussion, we comment that "[o]f note is very good performance of grammars achieved for CaMn despite a tiny size of the positive set (18 training sequences in each fold), which can be attributed to high homogeneity of the sample (50% identity on average)."

*5) The entire protein space is approximated by 829 single sequences of length 300-500, constructed from old data in 2006. Uniprot contains more than 100 million sequences, Pfam lists more than 16,000 protein families, the majority of them containing structurally resolved examples in the PDB. So I would guess that the negative set is an extremely rough approximation of protein sequence space, but the resulting sample is already three orders of magnitude larger than the corresponding npos values.*

We thank the Reviewer for the remark. We added a sentence in subsection "Materials" of "Basic evaluation" reporting that "[t]he ratio between negative and positive samples was high and varied from 1190:1 for CaMn to 207:1 for HET-s." Please note that in the newly added section "Sample applications", the motifs are searched versus more recent application-specific background sets.

*Comments for the Author*

*I am afraid that I miss some basic point, but even after reading the manuscript several times I continue to miss it. I would therefore suggest the authors to substantially revise their manuscript to make their work more accessible and the motivations and limitations – in particular for biological applications – more clear.*

We are thankful for the critical review of our manuscript. We are convinced that with the help of the comments, we significantly improved presentation of our work.

---

## Round 0.3 · accepted · Accept

Thank you for doing one more round of revisions.

#